# UniVerse-1: Unified Audio-Video Generation via Stitching of Experts

## Abstract

We introduce **UniVerse-1**, a unified, Veo3-like model capable of simultaneously generating coordinated audio and video. To enhance training efficiency, we bypass training from scratch and instead employ a **stitching of experts (SoE)** technique. This approach deeply fuses the corresponding blocks of pre-trained video and music generation experts models, thereby fully leveraging their foundational capabilities. To ensure accurate annotations and temporal alignment for both ambient sounds and speech with video content, we developed an **online annotation pipeline** that processes the required training data and generates labels during training process. This strategy circumvents the performance degradation often caused by misalignment text-based annotations. Through the synergy of these techniques, our model, after being finetuned on approximately $7,600$ hours of audio-video data, produces results with well-coordinated audio-visuals for ambient sounds generation and strong alignment for speech generation. To systematically evaluate our proposed method, we introduce **Verse-Bench**, a new benchmark dataset. In an effort to advance research in audio-video generation and to close the performance gap with state-of-the-art models such as Veo3, we will make our model and code publicly available. We hope this contribution will benefit the broader research community.

## 1 Introduction

The era of diffusion models (Song et al., 2020; Song & Ermon, 2020; Lipman et al., 2022) has culminated in the rise of Diffusion Transformer (DiT) architectures (Peebles & Xie, 2023), exemplified by landmark models like Sora (OpenAI, 2024) and its open-source counterparts (Kong et al., 2024; Yang et al., 2024; Wan et al., 2025). These models, leveraging unprecedented scales of data and computation, have achieved remarkable quality and prompt alignment in video generation. This success is catalyzing a profound transformation across creative industries and has spurred a wave of research into downstream applications such as talking head synthesis (Wang et al., 2023; Yu et al., 2023; Wang et al., 2025; Luo et al., 2025) and human animation (Tan et al., 2024; Chen et al., 2025; Lin et al., 2025), which now offer viable real-world solutions.

However, this rapid progress has been almost exclusively confined to the visual domain, treating video as a silent movie. This unimodal focus represents a fundamental bottleneck, as it ignores the inherently multimodal nature of video. Post-hoc video-to-audio models (Shan et al., 2025) serve as a superficial fix, but they are inherently limited; while capable of adapting audio to existing visual content, they fail to enforce temporal alignment in the reverse direction. This makes critical tasks, such as synchronizing lip movements with speech, impossible. While closed-source systems like Google's Veo3 (DeepMind, 2025.5) have demonstrated synchronous audio-video generation, the lack of publicly available technical details leaves a critical gap in open research.

To bridge this gap between closed-source systems and open research, we introduce **UniVerse-1**: a unified, fully open-source, Veo3-like model capable of simultaneously generating coordinated audio and video. Our work is underpinned by several key technical contributions designed to address the unique challenges of bimodal generation.

Instead of the costly process of training a new model from scratch, we propose a novel and efficient **stitching of experts (SoE)** paradigm. This methodology effectively fuses a state-of-the-art video generation model, WAN2.1 (Wan et al., 2025), with a music generation model, Ace-step (Gong et al.,

2025). The core of this fusion lies in lightweight, cross-modal MLP connectors introduced within corresponding blocks of each model. These connectors facilitate bidirectional interaction between modalities, and we found this strategy to significantly accelerate training convergence by leveraging the powerful priors of the pre-trained experts.

Furthermore, we tackle the critical challenge of data alignment in bimodal training. We argue that static, pre-processed annotations are a flawed paradigm for tasks requiring precise temporal consistency. To address this, we developed an **online annotation pipeline** that generates labels dynamically during training. This approach ensures strict temporal and semantic alignment between audio-video data and their textual descriptions, mitigating the performance degradation caused by static misalignment. During our investigation, we also uncovered a crucial, yet overlooked, factor in bimodal diffusion modeling: **cross-modal noise correlation**. We identified that the standard pseudo-random number generation process (Hamming, 1952) can introduce spurious correlations between the noise vectors for video and audio, which subsequently degrades audio quality during inference. Our solution involves ensuring independent noise sampling for each modality.

To support this work, we curated a high-quality dataset comprising approximately $7,600$ hours of precisely aligned audio-video content. To systematically evaluate our method, we also propose **Verse-Bench**, a new benchmark featuring 600 image-text prompt pairs covering a diverse range of sound categories. Highlighting its versatility, Verse-Bench supports not only joint audio-video generation but also unidirectional tasks, including a specialized **Verse-Ted** subset designed for evaluating audio-to-video synthesis.

In summary, our primary contributions are:

- **An Open-source Audio-Video Foundation Model:** We present UniVerse-1, a novel, open-source model capable of producing highly coherent and well-aligned synchronous audio-visual content, closing a critical gap in the open-source community.

- **A Novel Methodology for Joint Audio-Video Generation:** We propose a comprehensive methodology to enable efficient and high-quality joint audio-video synthesis. This is achieved through three key innovations: a **stitching of experts (SoE)** paradigm to accelerate convergence by fusing pre-trained models; an **online data annotation pipeline** to solve the critical static misalignment problem in training (Appendix. C for more details); and the identification and mitigation of the previously overlooked **cross-modal noise correlation** issue, a crucial factor for generation quality.

- **A Comprehensive Evaluation Benchmark:** We propose Verse-Bench, a new benchmark designed to comprehensively evaluate joint audio-video generation models across a diverse set of tasks.

## 2 RELATED WORKS

**Video Diffusion Models**   The field of video generation was revolutionized by the introduction of diffusion models, with pioneering works like AnimateDiff (Guo et al., 2023) and Video Diffusion Models (Ho et al., 2022) marking the beginning of this new era. This initial wave of research was further advanced by models such as Stable Video Diffusion (Blattmann et al., 2023), which first demonstrated that curating large-scale, high-quality datasets is critical for enhancing model performance. A common characteristic of these early models was their reliance on UNet architectures. To mitigate the challenges posed by the limited availability and quality of video data compared to images, these models were typically fine-tuned from pre-trained UNet image foundations. A significant paradigm shift occurred with the introduction of Sora (OpenAI, 2024), which heralded a new age defined by the Diffusion Transformer (DiT) architecture (Peebles & Xie, 2023) and training on massive, high-quality video corpora. This breakthrough spurred a proliferation of subsequent research. CogVideox (Yang et al., 2024) was the first to release an open-source DiT-based model, providing a significant catalyst for community-driven innovation. This was followed by other notable open-source models such as HunyuanVideo (Kong et al., 2024), WAN2.1 (Wan et al., 2025), and Step-Video (Ma et al., 2025), as well as high-performing closed-source systems including Kling (Kuaishou, 2024.06), SeeDance 1.0 (Gao et al., 2025b), Movie Gen (Polyak et al., 2024), and Veo2 (DeepMind, 2024.12). Architecturally, these contemporary models converge on a common blueprint. They employ a 3D Variational Autoencoder (VAE) to achieve spatio-temporal compression of video into a latent space. The core generative process is then handled by a DiT, which learns to denoise these noisy latents. Across these state-of-the-art models, the quality and scale of the training data have been

identified as paramount factors, making data curation and processing a central component of their development.

**Audio Diffusion Models**  The application of diffusion models to audio generation has followed a parallel trajectory to their video counterparts, fundamentally transforming the landscape of text-to-audio and text-to-music synthesis. Early explorations demonstrated the potential of diffusion for generating high-fidelity audio, but a pivotal advancement was the adoption of latent diffusion architectures (Rombach et al., 2022), which significantly improved both efficiency and quality. A common technical pipeline for these approaches involves first transforming the raw audio waveform into a mel spectrogram. A Variational Autoencoder (VAE) is then trained on this spectrogram representation to learn a compressed latent space, within which the core diffusion process operates. Within this framework, models such as Stable Audio Open (Evans et al., 2025) and Riffusion (Forsgren & Martiros., 2022) excel at generating high-fidelity, long-form audio. Furthermore, models like the AudioLDM series (Liu et al., 2024b; 2023) and DiffRhythm (Ning et al., 2025) advance these capabilities to vocal music synthesis, offering fine-grained control over rhythm and other expressive attributes.

**Joint Audio and Video Generation**  The exploration of joint audio-video generation within diffusion frameworks began with pioneering efforts like MM-Diffusion (Ruan et al., 2023). This model was the first to tackle this bimodal task, employing a UNet architecture with two distinct subnetworks, each dedicated to processing the audio and video modalities, respectively. Following this initial work, a series of subsequent models emerged (Ishii et al., 2024; Hayakawa et al., 2024; Ergasti et al., 2025). However, these early approaches were typically constrained by small-scale training datasets (Lee et al., 2022; Li et al., 2021), often less than 10 hours in size, which inherently limited their diversity and generalization capabilities. A notable step forward was made by models such as Syncflow (Liu et al., 2024a) and Uniform (Zhao et al., 2025), which scaled up the training data to approximately 500 hours by leveraging the VGGSound (Chen et al., 2020) and AudioSet (Gemmeke et al., 2017) dataset, thereby enhancing their generalization. Despite this progress, persistent challenges remained, including suboptimal video quality and a low degree of disentanglement between the audio and visual streams. The advent of Google's Veo3 (DeepMind, 2025.5) marked a significant milestone, representing the first large-scale initiative in synchronous audio-video generation. Veo3 demonstrated the capacity for generating high-fidelity audio and video that is not only diverse but also semantically and temporally coordinated, strictly adhering to user-provided text prompts.

## 3  UNIVERSE-1

### 3.1  PRELIMINARY

Our model is constructed upon the foundations of the Wan2.1 (1.3B parameters) (Wan et al., 2025) text-to-video model and the Ace-step (3.5B parameters) (Gong et al., 2025) music generation model. Before delving into technical details, we will briefly introduce their respective architectures.

**Wan2.1 model.** Wan2.1 is composed of three primary components: a 3D Variational Autoencoder (VAE), an umT5 (Chung et al., 2023) text encoder, and a Diffusion Transformer (DiT). The 3D VAE compresses an input video of shape $(3, T, H, W)$ into a latent representation of shape $(16, T/t, H/h, W/w)$, where the temporal and spatial downsampling factors are $t = 4$ and $h = w = 8$, respectively. Prior to being input to the DiT, this latent tensor is patchified using a kernel of $(1, 2, 2)$, and the resulting tokens serve as the input sequence. The umT5 model encodes the text prompt, and its embeddings are injected into the DiT via cross-attention to condition the generation. The model is trained to predict the velocity, which is used in the denoising step to recover the clean latent. Finally, the decoder of 3D VAE reconstructs this latent into the final video.

**Ace-step model.** Ace-step consists of a Music-DCAE (Deep Compression Autoencoder) (Chen et al., 2024), a umT5 text encoder, a lyric encoder, a speaker encoder, and a DiT. The raw audio waveform is first converted into a mel spectrogram. The Music-DCAE then encodes the input spectrogram of shape $(2, T, F)$ into a latent representation of shape $(8, T/t, F/f)$, with downsampling factors $t = f = 8$ along the temporal and frequency axes. This latent is subsequently patchified using a kernel of $(16, 1)$ to produce the input tokens for the DiT. Conditional control is provided by three sources: the umT5 encoder for the music style prompt, the lyric encoder for the lyrics, and the

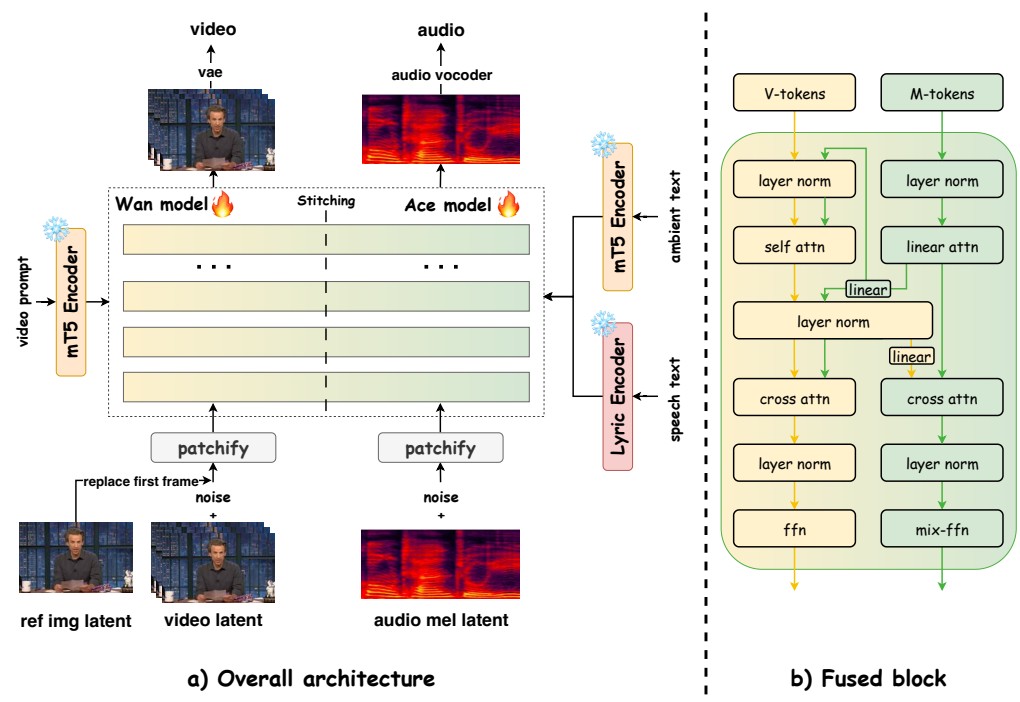

Figure 1: **Architecture of UniVerse-1.** (a) Overall architecture. The architectural foundation of UniVerse-1 is realized through a stitching of experts methodology. This approach deeply integrates the pre-trained Wan2.1 video model and the Ace-step audio model. (b) Fused block. The fusion is implemented at a granular, block-by-block level, where each block in the Wan architecture is deeply fused with its corresponding block in the Ace-step architecture.

speaker encoder for the speaker ID. These three embeddings are concatenated along the channel dimension and injected into the DiT via cross-attention. The model predicts the velocity to obtain the clean latent, which the Music-DCAE's decoder reconstructs into a mel spectrogram. A HiFiGAN vocoder (Liao et al., 2024) is then used to convert this spectrogram into the final audio waveform.

**Base model pre-training.** The learning objective for both of the aforementioned models is *Flow Matching* (Lipman et al., 2022). This fashion trains a neural network, $v_\Theta(\cdot, t, c)$, to predict a velocity field that transports samples from a simple source distribution, $p_0$ (*e.g.*, Gaussian noise), to a complex target data distribution, $p_1$. Specifically, these models leverage Conditional Flow Matching. Given a noise sample $x_0 \sim p_0$ and a data sample $x_1 \sim p_1$, a simple linear interpolation path is defined for time $t$:

$$x_t = (1-t)x_0 + tx_1. \tag{1}$$

The target velocity vector along this path is constant: $u_t = x_1 - x_0$. The model $v_\Theta(x_t, t, c)$, conditioned on $c$, is trained to predict this vector by minimizing the following L2 loss:

$$\mathcal{L}_{\text{FM}} = \mathbb{E}_{t \sim U(0,1), x_0 \sim p_0, x_1 \sim p_1} \left[ \| v_\theta((1-t)x_0 + tx_1, t, c) - (x_1 - x_0) \|^2 \right].$$

This objective directly trains the model to learn the vector field that maps noise to data, which leads to more stable and efficient training compared to traditional score-matching objectives.

### 3.2 METHOD

The overall architecture of our method is depicted in Fig. 1. We introduce several targeted modifications to the input stages of the original Wan2.1 and Ace-step models to facilitate bimodal integration and control. For the video component (Wan2.1), we enable conditioning on a reference image. During the forward process, the first frame of the noisy video latent is replaced with the corresponding clean latent representation of the provided reference image. For the audio component (Ace-step), we perform two adjustments. First, to ensure temporal alignment with the video's 25 frames-per-second

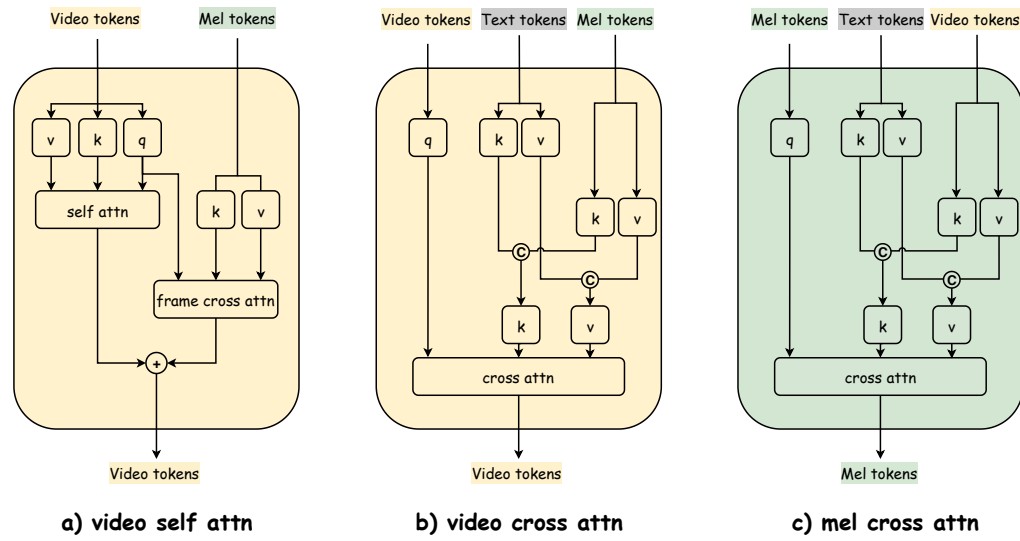

**Figure 2: Revised attention of UniVerse-1.** (a) Self attention of video branch, with additional mel tokens as input. (b) Cross attention of video branch, with additional mel tokens as input. (c) Cross attention of mel branch, with addition video tokens as input.

(fps) rate, the input mel spectrograms are processed at a 25.6 kHz sampling rate instead of the original 44.1 kHz. Second, to generalize the model beyond speaker-specific generation, we have removed the speaker encoder and its corresponding input from the architecture.

### 3.2.1 STITCHING OF EXPERTS

We introduce a novel framework, termed "Stitching of experts", for integrating specialized, pre-existing models for video and audio synthesis. This approach is designed to preserve the generative capabilities of each unimodal expert while simultaneously enabling fine-grained, bidirectional interaction between them at the level of individual layer blocks. To enhance training efficiency and leverage the powerful priors of these pre-trained models, we apply the stitching technique to the Wan2.1 and Ace-step models at the transformer block level. This process results in a unified, dual-stream architecture where each block co-processes information from both the video and audio modalities, functioning akin to a Mixture-of-Experts (MoE) layer.

As illustrated in Fig 1. b), we facilitate bidirectional cross-modal communication within each block. Specifically, the hidden states from the video stream, following its self-attention module, are injected into the audio stream's cross-attention module. Conversely, the hidden states from the audio stream, following its linear attention module, are reciprocally injected into the video stream's self-attention and cross-attention module. To ensure consistent feature scaling, the hidden states from both streams are jointly passed through a shared LayerNorm layer before injected into video stream's attention. Prior to injection, the cross-modal hidden states are passed through a two-layer linear adapter for feature space alignment. In the video stream, for instance, features from the audio branch are projected using dedicated key ($k_{proj}$) and value ($v_{proj}$) layers as shown in Fig. 2(a). A frame-by-frame cross-attention is then performed with the queries from the video stream to ensure alignment between the video and audio. Within each stream's respective cross-attention mechanism(shown in Fig. 2(b) and Fig. 2(c)), the conditioning signal from the other modality is projected using dedicated key ($k_{proj}$) and value ($v_{proj}$) layers. These new key-value pairs are then concatenated with the original text-derived key-value pairs along the context dimension, thereby enriching the conditioning information with cross-modal context.

### 3.2.2 LAYER INTERPOLATION

A key challenge in stitching the Wan2.1 and Ace-step models is the architectural mismatch in their depth, as they possess a different number of transformer blocks. To reconcile this disparity, we

introduce a **layer interpolation technique**. This method involves first calculating the difference in the number of layers. We then strategically insert new blocks at uniform intervals into the shallower of the two models until their depths align. Crucially, the parameters for each new block are initialized by linearly interpolating the weights of its immediately adjacent (bracketing) layers. This initialization strategy effectively bridges the architectural gap while ensuring a smooth performance trajectory during training, thereby mitigating the risk of training instability and severe performance oscillations.

### 3.2.3 TRAINING LOSS

In addition to the primary Flow Matching objective (Sec. 3.1), we incorporate two additional loss functions.

**Semantic Alignment Loss**  For the audio modality, we employ a **Semantic Similarity Loss** ($\mathcal{L}_{\text{SSL}}$), a technique consistent with Ace-step, to enhance the semantic fidelity of the generated audio. This loss operates by aligning an intermediate feature representation, $h_{\text{audio}}$, extracted from the audio stream of our fused block at a specific layer $L$ ($L = 12$ in our configuration). This internal representation is aligned against target features derived from two expert, pre-trained audio models:

- **MERT**(Music Encoder Representations from Transformers) (Li et al., 2023), which provides a general musical representation, $h_{\text{mert}}$, with a dimensionality of 1024 x $T_m$ (at a 75 Hz frame rate).
- **mHuBERT**(multilingual HuBERT) (Boito et al., 2024), which provides a speech-centric representation, $h_{\text{mHuBERT}}$, with a dimensionality of 768 x $T_h$ (at a 50 Hz frame rate).

To compute this loss, the intermediate feature $h_{\text{audio}}$ is first processed by two separate projection heads ($\pi_{\text{MERT}}$ and $\pi_{\text{mHuBERT}}$) and temporally interpolated to match the dimensionality and sequence length of $h_{\text{MERT}}$ and $h_{\text{mHuBERT}}$, respectively. The semantic similarity loss is then defined as the negative cosine similarity, encouraging the model's internal representations to align with those of the expert models:

$$\mathcal{L}_{\text{SSL}} = -\frac{1}{2}(\text{cosineSim}(h'_{\text{audio}}, h'_{\text{MERT}}) + \text{cosineSim}(h'_{\text{audio}}, h'_{\text{mHuBERT}}))$$

where $h'$ denotes the temporally aligned representations.

**Low Quality Data Loss Strategy**  The AudioSet and VGGSound datasets, while offering rich auditory diversity, are characterized by low visual fidelity. To leverage their strong audio content without corrupting the video generation quality, we employ a conditional loss scheme. Specifically, the Flow Matching loss for the video modality ($\mathcal{L}_{\text{FM-video}}$) is only computed for samples originating from these two datasets when the diffusion timestep $t$ exceeds an empirically determined threshold. We set this threshold to $\tau = 800$ (out of 1000 total timesteps). This strategy is predicated on the principle that at high noise levels (i.e., for $t > \tau$), the model learns to capture coarse, low-frequency features of the video, such as general motion and structure, which are less affected by the poor visual quality. By excluding the loss calculation at lower noise levels, we prevent the model from overfitting to the high-frequency visual artifacts and noise present in these datasets. The loss for video modality is:

$$\mathcal{L}_{\text{FM-video}} = \begin{cases} \mathcal{L}_{\text{FM}}(x_1, x_0, t) & if\ \ x_1 \in \Theta\ \ or\ \ (x_1 \in \zeta\ \ and\ \ t > 800) \\ 0 & else \end{cases}$$

where $x_0 \sim p_0$ is noise sample, $x_1 \sim p_1$ is data sample, $\zeta$ is data subset include vggSound and audioset, $\Theta$ is data subset exclude vggSound and audioset. The final training objective is a weighted sum of the flow matching and semantic alignment losses:

$$\mathcal{L} = \mathcal{L}_{\text{FM-video}} + \mathcal{L}_{\text{FM-mel}} + \lambda_{SSL} \cdot \mathcal{L}_{\text{SSL}}$$

where $\mathcal{L}_{\text{SSL}}$ is a hyperparameter controlling the influence of the SSL guidance, empirically set to 1.0 according to Ace-step.

### 3.3 INDEPENDENT NOISE SAMPLING STRATEGY

Our empirical investigation reveals a critical sensitivity of multi-modal diffusion models to the pseudo-random number generation process. When a single, fixed random seed is used to initialize a training run, the noise tensors for the video ($\epsilon_v$) and audio ($\epsilon_a$) modalities are sampled sequentially

from the same deterministic PRNG sequence. Due to the deterministic nature of the underlying algorithm (Linear Congruential method), this sequential sampling introduces a spurious structural correlation between the two noise tensors, which can be expressed as $\epsilon_a = f(\epsilon_v)$. This violates the critical assumption that the noise vectors are statistically independent.

The model inadvertently learns this spurious correlation as a shortcut during training. Consequently, during inference, any alteration to the sampling of $\epsilon_v$, such as a change in video resolution or duration, propagates through the PRNG's state and alters the structure of the subsequently sampled $\epsilon_a$. This mismatch with the learned correlation results in a significant degradation of the audio generation quality.

To address this, we propose an **Independent Noise Sampling Strategy**. This approach isolates the noise generation for each modality by employing separate and independently seeded PRNG instances. This method effectively breaks the deterministic correlation, ensuring the noise vectors are statistically independent. As a result, the model becomes robust to variations in inference-time conditions, mitigating the issue of performance degradation.

## 4 EXPERIMENTS

### 4.1 SETUP

**Implementation Details**  The training is conducted with an effective batch size of $128$ over $50k$ steps on a $7,600$-hour audio-visual datasets built by our data curation pipeline (Appendix. B for more details), using the AdamW optimizer with a learning rate of $5e-6$. We employ Fully Sharded Data Parallel (FSDP) for distributed training across multiple nodes, with a gradient accumulation step of $4$.

**Compared Methods**  We conduct a comprehensive evaluation of our model by benchmarking it against a suite of state-of-the-art baselines across several distinct generation tasks, the details are in Appendix. D. It is important to note that our model is constructed by stitching the pre-trained WAN2.1 (1.3B) and Step-Ace (3.5B) models. Consequently, the comparisons against the state-of-the-art baselines are intended to provide a qualitative reference and situate our work, rather than to make a direct claim of superior performance.

**Benchmark**  To construct our evaluation set, we curated $600$ image-text prompt pairs from a multitude of sources. These sources encompass frames extracted from YouTube videos, BiliBili videos, TikTok clips, movies, and anime; images generated by AI models (ByteDance, 2025; Wu et al., 2025); and a collection of images from public websites. More details are in Appendix. A.1

**Evaluation Protocol**  We quantitatively evaluate our method against baselines on the Verse-Bench benchmark. The evaluation is structured across 6 distinct generation tasks, each with a tailored set of metrics. More details are in Appendix. E.1.

To provide a holistic, quantitative comparison of joint audio-video generation capabilities, we introduce a composite **Overall Score**. This score is formulated as a weighted average of four sub-scores, reflecting different aspects of model performance:

$$Overall_{Score} = 0.5 * S_{joint} + 0.2 * S_{video} + 0.2 * S_{audio} + 0.1 * S_{other}$$

Reflecting our emphasis on the core task, the *Joint Quality Score* ($S_{joint}$) constitutes 50% of the total weight. To strongly couple audio-video temporal alignment (AV-A) with audio-text semantic consistency (CS), we compute $S_{joint}$ using the harmonic mean of their normalized values. This ensures that a high score is achieved only when both metrics are strong, heavily penalizing models that excel at synchronization but fail on content relevance. The remaining scores for video ($S_{video}$), audio ($S_{audio}$), and other modalities ($S_{other}$) are the arithmetic means of their respective metrics.

To ensure fair aggregation across metrics with different scales and trends, all individual metrics are first normalized to a unified range where higher is always better. For metrics with an upward trend ($\uparrow$), we use standard min-max scaling: (value - worst) / (best - worst). For metrics with a downward trend ($\downarrow$), we use an inverted formula: (worst - value) / (worst - best). More details are in Appendix. E.2.

Table 1: Quantitative results on Verse-Bench. Best results are in **bold**, second best are underlined.*When computing overall score for SVG, we set WER to 1.0 and LSE-C to 0.0.

| | Methods | Video | | Audio | | | | | | | TTS | Audio-Video | | Overall |
|---|---|---|---|---|---|---|---|---|---|---|---|---|---|---|
| | | AS↑ | ID↑ | FD↓ | KL↓ | CS↑ | CE↑ | CU↑ | PC↓ | PQ↑ | WER↓ | LSE-C↑ | AV-A↓ | Score↑ |
| Video Methods | CogVideox1.5 5B | 0.44 | 0.83 | - | - | - | - | - | - | - | - | - | - | - |
| | Wan2.2-14B | **0.50** | 0.88 | - | - | - | - | - | - | - | - | - | - | - |
| | Kling2.1 | 0.41 | 0.85 | - | - | - | - | - | - | - | - | - | - | - |
| | SeeDance1.0 | 0.47 | 0.86 | - | - | - | - | - | - | - | - | - | - | - |
| Audio Methods | Stable Audio | - | - | 1.13 | 1.38 | 0.28 | 3.88 | **6.15** | 2.60 | 6.53 | - | - | - | - |
| | AudioLdm2 | - | - | 1.21 | 2.30 | 0.24 | 3.98 | 5.88 | 3.43 | 6.04 | - | - | - | - |
| TTS Methods | Cosyvoice | - | - | - | - | - | - | - | - | - | 0.17 | - | - | - |
| | Cosyvoice2 | - | - | - | - | - | - | - | - | - | 0.16 | - | - | - |
| | VibeVoice | - | - | - | - | - | - | - | - | - | **0.15** | - | - | - |
| A2V Methods | Fantasy-Talking | 0.42 | 0.87 | - | - | - | - | - | - | - | - | 2.68 | - | - |
| | Wan-S2V | 0.46 | **0.89** | - | - | - | - | - | - | - | - | **6.49** | - | - |
| V2A Method | HunyuanVideo-Foley | - | - | **0.82** | **1.27** | **0.40** | **4.04** | 5.72 | 3.09 | 6.27 | - | - | 0.78 | - |
| Joint Methods | SVG | 0.41 | 0.25 | 1.55 | 3.62 | 0.08 | 2.93 | 5.50 | **2.35** | 6.26 | - | - | **0.09** | 0.220* |
| | Ovi | 0.52 | 0.89 | - | - | 0.09 | 4.89 | 6.04 | 2.20 | 6.23 | 0.28 | 4.85 | 0.51 | 0.408 |
| | **Ours** | 0.47 | **0.89** | 1.25 | 2.70 | 0.16 | 3.53 | 4.61 | 2.49 | 5.20 | 0.18 | 1.34 | 0.23 | **0.403** |

## 4.2 QUANTITATIVE EVALUATION

We compare Universe-1 against a range of state-of-the-art (SOTA) models specialized for specific tasks as shown in Tab. 1. As a unified joint generation model, a direct comparison of single-modality metrics against these "expert" models presents inherent complexities. Nevertheless, our model demonstrates robust capabilities across multiple dimensions.

In terms of video quality, Universe-1 achieves the highest score in identity preservation (ID: 0.89), showcasing its superior ability to maintain subject consistency throughout the generation process. For audio quality, while there is a gap compared to leading audio-only generation models, our model obtains a highly competitive score in pitch correlation (PC: 2.49).

The core strength of our model lies in synchronous audio-video generation. It is crucial to interpret the metrics for such joint generation tasks with caution. For instance, in the video-to-audio (V2A) setting, the AV-A metric for a model like Hunyuanvideo-Foley is calculated with a ground-truth video, whereas our model generates both modalities simultaneously. Therefore, AV-A must be considered in conjunction with the audio-text CLAP score (CS) for a holistic assessment. From this integrated perspective, our model (AV-A: 0.23, CS: 0.16) demonstrates a better overall audio-visual content consistency than SVG (AV-A: 0.09, CS: 0.08).

Similarly, for the lip-sync (LSE-C) metric, our model's score (1.34) is evaluated on fully generated audio and video, making it susceptible to the quality of both generated modalities. In contrast, audio-to-video (A2V) methods like Wan-S2V are evaluated using ground-truth audio, which naturally yields a higher score (6.49). Despite this evaluation disparity, Universe-1 achieves promising results as the first open-source joint generation framework of its kind, establishing a solid foundation for future research.

To encapsulate the model's holistic performance across these diverse and complex trade-offs, we designed a composite Overall Score detailed in the appendix. Ultimately, Universe-1 achieves a superior Overall Score of 0.403, substantially outperforming the SVG baseline (0.220). This result quantitatively validates the effectiveness of our approach and establishes Universe-1 as a state-of-the-art, well-balanced joint generation model.

## 4.3 USER STUDY

We conducted a user study, the details and results of which are presented in Appendix. F.

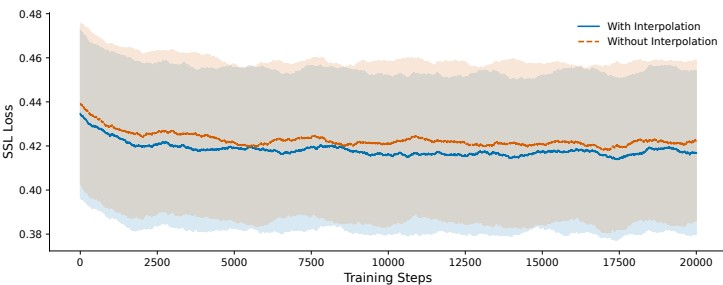

Figure 3: **Ablation study on Linearly Interpolating.** SSL Loss with(blue) and without(red) Linearly Interpolating.

Table 2: Ablation study on the effectiveness of our proposed components. The Overall Score is a composite metric detailed in the appendix. Removing either component degrades performance, validating their contribution.

| Method | Video | | Audio | | | | | | | TTS | Audio-Video | | Overall |
|---|---|---|---|---|---|---|---|---|---|---|---|---|---|
| | AS ↑ | ID ↑ | FD ↓ | KL ↓ | CS ↑ | CE ↑ | CU ↑ | PC ↓ | PQ ↑ | WER ↓ | LSE-C ↑ | AV-A ↓ | Score ↑ |
| w/o LQLS | 0.44 | 0.78 | 1.26 | 2.84 | 0.15 | 3.38 | 4.35 | 2.58 | 4.97 | **0.16** | **1.35** | 0.28 | 0.378 |
| w/o INSS | 0.43 | 0.75 | 1.43 | 3.51 | 0.11 | 2.44 | 3.14 | 2.92 | 3.99 | 0.38 | 0.99 | **0.18** | 0.316 |
| **Ours** | **0.47** | **0.89** | **1.25** | **2.70** | **0.16** | **3.53** | **4.61** | **2.49** | **5.20** | 0.18 | 1.34 | 0.23 | **0.403** |

## 4.4 ABLATION STUDY

We performed an ablation study to investigate the contributions of our Low Quality data Loss Strategy(LQLS) and Independent Noise Sampling Strategy(INSS), as shown in Tab. 2. The findings indicate that LQLS provides improvement accross video quality and consistency ID, thus confirming its efficacy and positive impact on training. Furthermore, the results for INSS demonstrate a significant enhancement in audio generation quality, validating the effectiveness of this approach.

We also conducted an ablation study to validate the effectiveness of initializing parameters via linear interpolation within our layer interpolation technique. We provide a visual comparison of the semantic alignment loss, as shown in the Fig. 3. The results indicate that with interpolated initialization, the model exhibits a superior capacity for continued learning. Compared to randomly initializing the new layers, our approach aligns with the correct audio semantics much earlier in the training process.

## 5 LIMITATION AND FUTURE WORK

The work presented herein constitutes an initial exploration into unified audio-video generation. Our study was constrained by computational resources, necessitating that we conduct training exclusively on the Wan2.1-1.3B video model. The performance of our model is, therefore, inherently limited by the capacity of this base model. In the future, our research will focus on two key directions. First, we will scale up our experiments to larger video foundation models. Second, we will engage in more extensive and refined data curation efforts. The ultimate objective is to significantly advance the capabilities of open-source audio-video synthesis models, thereby bridging the performance gap to state-of-the-art proprietary models.

## 6 CONCLUSION

In this paper, we presented UniVerse-1, a novel framework for joint audio-video synthesis, achieved through the deep integration of a video foundation model and a music generation model using stitching of experts. Following fine-tuning on the dataset we curated, we also introduced Verse-Bench, a comprehensive benchmark to foster comparative research. To promote reproducibility and further innovation, our model and code have been made publicly available. Our experimental results validate

that this methodology offers a viable and efficient pathway for building sophisticated multimodal generative models by leveraging pre-existing unimodal foundations.

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

APPENDIX

## A  MORE DETAILS ABOUT VERSE-BENCH

### A.1  DETAILED DESCRIPTION

Our dataset comprises three subsets.

- Set1-I contains image-text pairs (including AI-generated, web-crawled, and media screenshots), for which video/audio captions and speech content were produced using LLMs (Xu et al., 2025) and manual annotation, comprising a total of 205 samples. Statistical results in 6.
- Set2-V consists of video clips from YouTube and Bilibili, which were annotated with LLM-generated (Xu et al., 2025) captions and Whisper-based ASR (Radford et al., 2023) transcripts, followed by human verification, comprising a total of 295 samples. Statistical results in 7.
- Set3-Ted (the Verse-Ted subset) includes TED Talks from September 2025, processed with the same annotation pipeline as Set2, comprising a total of 100 samples.

Our collected test data is highly diverse, encompassing a wide spectrum of audio categories: human speech; animal vocalizations (e.g., bird chirping, cat meowing); instrumental music (e.g., piano, guitar); natural sounds (e.g., thunder, rain); human-object interactions (e.g., keyboard typing, cooking, chopping vegetables); object-object interactions (e.g., glass shattering, marbles dropping); mechanical sounds (e.g., trains, airplanes), and so on. The complete statistical results of set1 and set2 are shown in 5.

### A.2  STATISTICAL RESULTS

This section provides the detailed category catalog for Verse-Bench, along with a high-resolution pie chart illustrating its statistical distribution in Fig. 5 6 7.

Category list is in Tab. 3, 4, 5.

## B  DATA CURATION

We curated a large-scale, high-quality dataset from a diverse range of sources to train our model. The primary component was sourced from YouTube, encompassing content such as music variety shows, classical music performances, cooking tutorials, public speeches, interviews, vlogs, and demonstrations of tool usage. This was supplemented with cinematic movie clips and high-quality stock footage from Pexels. To further bolster the audio modality, we also incorporated the widely-used VGGSound and AudioSet datasets.

For our self-collected data (YouTube, Pexels, and movie clips), we implemented a rigorous multi-stage filtering pipeline to ensure data quality and relevance:

- **Audio-Visual Pre-screening** Videos lacking an audio track were immediately discarded.
- **Quality Control** We filtered out content based on technical specifications: resolution below $1080p$, a bitrate-to-resolution ratio under $600$, and a DOVER (Wu et al., 2023a) aesthetic quality score below $0.6$.
- **Temporal Coherence** PySceneDetect[1] was applied to segment videos, and any resulting clip shorter than 5 seconds was removed to ensure meaningful duration.
- **Audio Activity Detection** To eliminate silent segments, we analyzed each audio track for metrics such as volume, energy, and zero-crossing rate.
- **Speech Content Verification** Whisper (Radford et al., 2023) was used to detect the presence of human speech. Clips without speech were retained as general audio-visual data. If speech was present, it proceeded to the next step.

---

[1]https://github.com/Breakthrough/PySceneDetect

- **Human Face Detection** For clips identified as containing speech, a second verification step was performed: we detected for the presence of a human face (Deng et al., 2020). If no face was found, the clip was discarded. If a face was present, we employed SyncNet (Chung & Zisserman, 2016) to verify the audio-visual correspondence (lip-sync). Only clips with a SyncNet confidence score above a threshold of 2.0 were retained and explicitly labeled as containing speech content.

For the VGGSound and AudioSet data, a simplified process was used where we either performed scene detection or segmented clips based on their existing timestamps, retaining only those longer than 5 seconds.

Following this comprehensive curation process, our final dataset comprises 7,685 hours of data. This is categorized into three subsets: 1,187 hours of verified speech-centric content, 3,074 hours of general-purpose audio-video data, and 3,422 hours from VGGSound and AudioSet primarily used for bolstering audio-specific training.

## C  ONLINE DATA ANNOTATION

Conventional offline annotation methods, where captions are pre-generated for entire videos, present a significant challenge for training generative models. During training, fixed-length clips are randomly sampled from these videos, often creating a temporal and semantic misalignment between the sampled clip and the global, pre-existing caption. This issue is particularly acute in the context of joint audio-video generation, where the temporal synchronization between an acoustic event and its description is critical. Even minor temporal shifts can render an audio annotation invalid.

To overcome these limitations, we propose and implement an Online Data Annotation Pipeline. This pipeline operates as a dedicated server process that runs concurrently with training. It dynamically fetches raw video, processes clips in real-time to generate precisely aligned data-annotation pairs, and populates a shared buffer. The main training process then acts as a consumer, fetching these ready-to-use data tuples, ensuring that every training instance is perfectly synchronized.

The online processing for each data tuple involves the following steps:

- **Temporal Sampling** A fixed-length segment (e.g., 5 seconds) is randomly extracted from a source video, yielding corresponding video and audio streams.

- **Multi-modal Annotation** The extracted audio-video clip is immediately passed to our annotation module for captioning.

- **Text and Video Encoding** The generated text prompts (for video, audio, and speech) are encoded. Concurrently, the video clip is encoded into a spatio-temporal latent representation using the 3D VAE.

- **Audio Encoding** The audio stream is converted to a mel spectrogram and subsequently encoded into a latent representation by the Music-DCAE (Chen et al., 2024).

The core of our pipeline is the multi-modal annotation step (Step 2), which proceeds as follows:

- **Speech Transcription** Whisper (Radford et al., 2023) is employed to perform Automatic Speech Recognition (ASR) on the sampled audio, yielding the raw speech content.

- **Structured Multimodal Captioning** We construct a structured prompt that incorporates the transcribed speech. This prompt, along with the audio and video streams of the clip, is fed into the QWen2.5-Omni (Xu et al., 2025) multimodal model. QWen2.5-Omni is specifically instructed to output three distinct, aligned annotations for the clip: the verified speech content, a descriptive video caption, and a caption for the ambient audio.

This online, just-in-time process guarantees that every training instance consists of video and audio latents that are perfectly synchronized in time and semantically consistent with their corresponding textual annotations, thereby eliminating the data misalignment problem inherent in offline methods.

# D DETAILS ON COMPARED METHODS

We conduct a comprehensive evaluation of our model by benchmarking it against a suite of state-of-the-art baselines across several distinct generation tasks:

- Video Generation: We compare our model against leading text-to-video systems, including Wan2.2 (14B) (Wan et al., 2025), HunyuanVideo (Kong et al., 2024), CogVideoX-1.5, (5B) (Yang et al., 2024), Kling 2.1 (Kuaishou, 2024.06), and SeeDance 1.0 (Gao et al., 2025b).

- Audio Generation: For the audio modality, we benchmark against established text-to-audio models such as Stable Audio Open (Evans et al., 2025) and AudioLDM2 (Liu et al., 2024b).

- Text-to-Speech (TTS): As a supplementary evaluation of vocal synthesis, we compare our model's performance against specialized TTS models, including CosyVoice (Du et al., 2024a), CosyVoice2 (Du et al., 2024b), and VibeVoice (Peng et al., 2025).

- Audio-to-Video methods: We also compare out model with talking-based audio to video methods such as FantastyTalking (Wang et al., 2025) and Wan-S2V (Gao et al., 2025a).

- Video-to-Audio methods: As a closely related task to joint audio-visual generation, we also benchmarked our model on the video-to-audio (V2A) task, drawing comparisons with state-of-the-art methods such as HunyuanVideo-Foley (Shan et al., 2025).

- Joint Audio-Video Generation: For our core task of synchronous audio-visual synthesis, we compare our method against existing prompt-based joint generation models: SVG (Ishii et al., 2024). Since methods such as MM-Diffusion (Ruan et al., 2023) and R-FLAV (Ergasti et al., 2025) are class-conditional generative models, a direct comparison with our approach is not applicable, we only compare with SVG here.

# E DETAILS ON EVALUATION PROTOCOL

## E.1 TASKS AND METRICS

The compared task and corresponding metrics are:

- Video Generation: Performance is assessed on three criteria:
  - Aesthetic Score (AS): This is a composite score averaging three components: fidelity, measured by MANIQA (Yang et al., 2022) to penalize blur and artifacts, and aesthetic quality, evaluated by both aesthetic-predictor-v2-5 (discus0434, 2024) and Musiq (Ke et al., 2021).
  - ID Consistency (ID): To measure identity preservation, we compute the mean DINOV3 (Siméoni et al., 2025) feature similarity between the reference image and each generated frame.

- Audio Generation: We evaluate audio quality from three perspectives:
  - Distributional Similarity: We measure the Fréchet Distance (FD) and Kullback-Leibler (KL) divergence between the generated and real data distributions, using features extracted from PANNs (Kong et al., 2020) and PaSST (Koutini et al., 2021).
  - Semantic Consistency: The alignment between the audio and the input text is measured by the LAION-CLAP (Wu et al., 2023b) score.
  - Quality and Diversity: We report the Inception Score (IS) calculated with a PANNs classifier. Additionally, we use AudioBox-Aesthetics (Tjandra et al., 2025) to assess Production Quality (PQ), Production Complexity (PC), Content Enjoyment (CE), and Content Usefulness (CU).

- Text-to-Speech (TTS): We evaluate synthesis accuracy using the Word Error Rate (WER), which is derived by transcribing the generated audio with the Whisper-large-v3 model (Radford et al., 2023).

- Audio-to-Video: We evaluate this task using the same criteria as the video generation task, additionally providing a SyncNet (Chung & Zisserman, 2016) confidence score to assess lip-sync accuracy.

- Video-to-Audio: This task use all metrics from audio generation tasks. Furthermore, we introduce the Audio-Video Alignment (AV-A) metric to specifically quantify the temporal synchronization

between the generated audio and video streams, which is computed via Synchformer (Iashin et al., 2024).

- Joint Audio-Video Generation: For this task, we use all relevant metrics from the individual tasks above.

The evaluation of our models and their components is conducted across the three test sets as follows:

- The video generation models and SVG are evaluated on Set 1 and Set 2.
- For the audio generation model is evaluated on Set1 and Set2.
- The Text-to-Speech (TTS) model is primarily evaluated on Set 3.
- The Audio-to-Video (A2V) model is evaluated exclusively on Set 3, while the Video-to-Audio (V2A) model is evaluated exclusively on Set 2.
- Finally, our complete Universe-1 model is benchmarked against all three test sets.

For audio generation, we evaluate the metrics CE, CU, PC, and PQ on Set 1. On Set 2, the evaluation is expanded to include FD, KL, and CS in addition to the aforementioned metrics. Furthermore, LSE-C is evaluated exclusively on Set 3, while the AV-A metric is also applied to Set 1 when evaluating UniVerse-1 and SVG.

### E.2   DETAILS ABOUT OVERALL SCORE

To provide a comprehensive and principled evaluation of model performance across multiple modalities, we designed a composite Overall Score. The primary motivation behind this metric is to create a single, holistic figure of merit that encapsulates the complex trade-offs inherent in joint audio-video generation, while placing a strong emphasis on the model's core capabilities. The score is formulated as a weighted average of four distinct sub-scores:

$$Overall_{Score} = 0.5 * S_{joint} + 0.2 * S_{video} + 0.2 * S_{audio} + 0.1 * S_{other}$$

The components are defined as follows:

- Joint Quality Score ($S_{joint}$): This is the most critical component, receiving a $50\%$ weight to reflect the primary focus of our work. Its purpose is to measure the quality of synchronous audio-video generation. A key insight is that temporal alignment (measured by AV-A) is only meaningful if the generated audio content is semantically relevant to the text prompt (measured by CS). To strongly couple these two aspects, we employ the harmonic mean of their normalized values. The harmonic mean is highly sensitive to lower values, ensuring a high score is achieved only when both alignment and content relevance are strong. A model with perfect alignment but irrelevant content, or vice-versa, will be heavily penalized.
- Video and Audio Quality Scores ($S_{video}$, $S_{audio}$): These sub-scores, each weighted at $20\%$, provide a balanced view of the model's capabilities in generating high-quality content for each individual modality. They are calculated as the arithmetic mean of all normalized metrics within their respective categories.
- Other Modalities Score ($S_{other}$): This component, with a $10\%$ weight, accounts for secondary capabilities such as Text-to-Speech (WER) and audio-driven lip-sync (LSE-C), ensuring the model's versatility is also recognized.

**Metric Normalization**   To ensure fair aggregation of metrics with different scales and trends (i.e., some are better when higher, others when lower), all individual metrics are first normalized to a unified [0, 1] scale, where a higher score is always better. We use a robust normalization scheme based on fixed theoretical or empirical bounds, rather than the min-max values within our specific results table, to ensure the stability and generalizability of the score.

For upward-trending metrics ($\uparrow$), such as AS or CE, we use the formula:

$$Normalized_{Score} = (value - worst_{bound}) / (best_{bound} - worst_{bound})$$

For downward-trending metrics ($\downarrow$), such as FD or AV-A, we use an inverted formula to reverse the trend:

$$Normalized_{Score} = (worst_{bound} - value) / (worst_{bound} - best_{bound})$$

For metrics with a natural scale, such as the [1, 10] range for CE, CU, PQ, and PC, we use the scale's limits as the best bound and worst bound. For unbounded metrics like FD and KL, we set a conservative empirical upper bound as the worst bound (e.g., 3.0 and 4.0, respectively) and their theoretical optimum of 0 as the best bound.

**Handling of Missing Data**   For models with missing metrics, such as the SVG baseline which lacks scores for WER and LSE-C, we performed imputation by assigning the worst possible values (e.g., WER=1.0, LSE-C=0.0). This conservative approach results in a normalized score of 0 for these imputed metrics, allowing for a direct and fair comparison without artificially inflating the baseline's performance.

This carefully designed Overall Score provides a robust, interpretable, and single-figure metric that accurately reflects the strengths of a well-balanced, joint audio-video generation model.

## F   USER STUDY

To provide a comprehensive qualitative assessment of our model's capabilities, we conducted a user study comparing our method against state-of-the-art specialized models across six distinct generation tasks. The results, presented as user preference percentages in Fig. 4, are detailed below.

### F.1   DETAILS

Our user study was conducted with 17 participants, who were asked to independently evaluate the results from various methods across different tasks. For each task dimension, 10 comparison sets were provided. Following a standardized evaluation protocol, participants were instructed to rank the outputs of the different methods without allowing for ties. After collecting the rankings, we calculated the mean rank for each method to quantify the collective user preference. These mean ranks were subsequently normalized into percentages for final presentation.

### F.2   RESULTS

**Key Findings**   Our model, demonstrates a powerful and well-balanced performance profile as a unified audio-video framework. Our model not only establishes a dominant lead in the core joint generation task but also achieves SOTA or highly competitive results in complex directional tasks like Text-to-Speech (TTS) and Audio-to-Video (A2V). The study validates that our *Stitching of Experts* approach creates a versatile model that excels where multi-modal coherence is paramount.

**Task-Specific Analysis**

- **(a) Video-Only and (c) Audio-Only Generation:** In unimodal tasks, our model is predictably surpassed by larger expert models. This is a direct reflection of using a more modest video foundation, the Wan2.1 1.3B model, within a unified architecture, which naturally has a lower intrinsic capability than the specialized SOTA systems.

- **(b) Video-to-Audio (V2A) Generation:** The expert model, HunyuanVideo-Foley, achieves a higher user preference in this specialized directional task.

- **(e) Text-to-Speech (TTS) Generation:** Our model demonstrates a significant strength in text-to-speech synthesis. In the Text Alignment metric, it outperforms both CosyVoice and CosyVoice2, proving its ability to generate high-quality, semantically aligned audio from text is highly competitive with SOTA TTS models.

- **(f) Audio-to-Video (A2V) Generation:** The A2V results reveal a key advantage of our approach. While a specialized model leads in Lip Sync, our model overwhelmingly outperforms all baselines in Reference Consistency. This is a critical finding, indicating our model's superior capability in maintaining the subject's identity and scene integrity, a vital component for realistic and coherent video synthesis that specialized models often overlook.

- **(d) Joint Audio-Video Generation:** This task provides the most holistic evaluation and represents the core strength of our work. Our model demonstrates a dominant, landslide victory over the SVG baseline across all five metrics. This substantial margin unequivocally establishes the superiority of

our framework in the challenging task of simultaneous and coherent audio-video synthesis from a shared latent space.

**Overall Conclusion from User Study**   The user study provides a clear and compelling validation of our unified framework. Ours distinguishes itself not as a master of one trade, but as a master of a far more complex one: multi-modal coherence. It establishes state-of-the-art performance in simultaneous audio-video generation, demonstrates a critical and superior capability in A2V reference consistency, and is highly competitive in TTS text alignment. While unimodal expert models lead in their narrow domains, our model's success across the most challenging joint and directional tasks proves that our approach effectively pioneers a versatile, powerful, and efficient paradigm for open-source audio-video joint generation.

## G   QWEN2.5-OMNI PROMPT USED IN DATA PIPELINE

---

**Prompt1**

```
You are a video marking expert responsible for labeling the provided
video and audio and outputting JSON. The main labels include:
```

1. The overall detailed style description of the video, such as realistic, abstract, cartoon, freehand, 1960s, etc.

2. Characters appearing in the video or well-known IPs (Iron Man, Minions, etc.), with detailed appearance descriptions, such as a man wearing a black suit, a white shirt inside the suit, a red tie, and black narrow-framed glasses, etc.

3. Objects appearing in the video, with detailed appearance descriptions, such as a football with white, red, and blue stripes in the lower left corner.

4. Interactions between characters and objects or between characters, such as a man walking towards the football and trying to pad it.

5. Background description of the video, such as a light blue sea in the background, a white beach in front, and some colorful balloons scattered on the beach.

6. Character role in the video and their speech content, such as [the man saying: 'Today we are going to learn beach football, which is completely different from beach volleyball.']

7. Dynamic description of events in the video, such as when the man lifts the football with his foot, the football is kicked high and flies to the right of the screen, the screen moves to the right with the football, and the football falls into the basket on the beach.

8. Ambient sounds in the video, such as the sound of seagulls, the whistle of ships, the sound of waves.

9. Comprehensive description, objectively and detailedly describing the content of the video, emphasizing the main content of the video, and describing the character movements, camera switching methods, background environment, ambient sounds and speech content in the video audio, etc.

**Instruction:**  You should first infer how many speakers are in the video implicitly, then identify which character said what content accurately according to video lip movement and corresponding audio speech; use the ASR results as a reference for speech content recognition.

**prompt2**

```
[Output requirements]:  The above content is used as a basic unit,
and multiple unit contents are described cyclically as the video
progresses.  Each item in the output dict should include a detailed
description respectively, and the content is output in JSON format.
The basic format is as follows:
------Example Start------
{
   "Video Style":  "xxx",
   "Unit 1":  {
     "Character":  ["xxx", "xxx"],
     "Character descriptions":  ["xxx", "xxx"],
     "Object":  ["xxx", "xxx", "xxx"],
     "Character Interaction":  "xxx",
     "Background Description":  "xxx",
     "Speech Content":  ["xxx", "xxx"],
     "Event Dynamic Description":  "xxx",
     "Ambient Sound":  ["xxx", "xxx"],
     "Comprehensive Description":  "xxx"
   },
   ...
   "Unit n":  {
     ...
   },
}
------Example End------
[Note]:  Please fill in the above content according to the video
content, ignore the possible subtitles in the video.  The output
format is JSON.
```

The complete prompt to Qwen2.5-Omni is $Prompt_1 + transcript + Prompt_2$. Where $transcript$ is the tts results from Whisper.

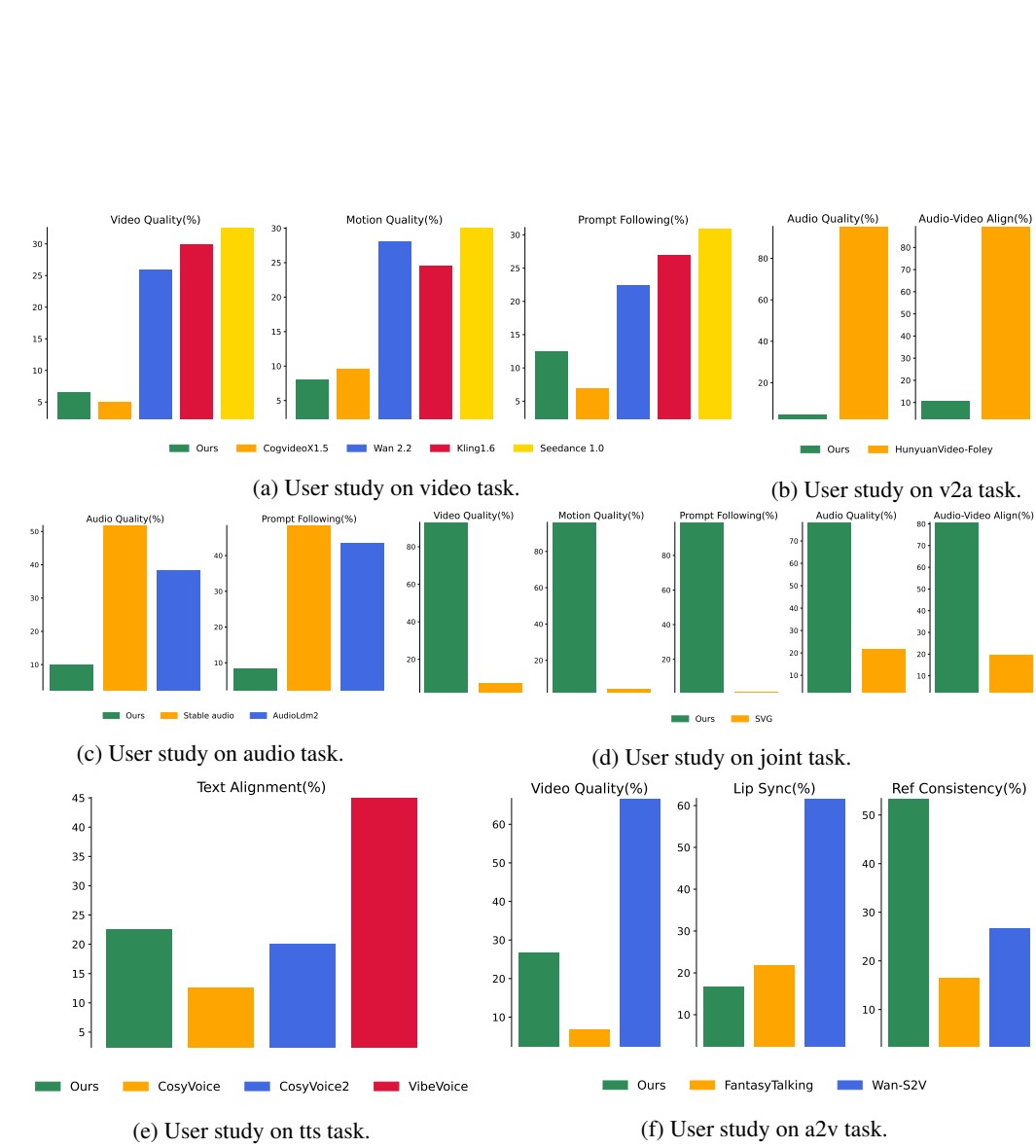

Figure 4: User study of our method with baseline methods. Best viewed with **zoom-in**.

Table 3: Detailed Audio Classification Statistics for Set 1

| Category / Subcategory | Count | Percentage (%) |
|---|---|---|
| **Natural Environment** | **70** | **36.3** |
| Weather-Wind | 18 | |
| Plants-Vegetation | 12 | |
| Animals-Birds | 10 | |
| Water-Ocean/Waves | 8 | |
| *Other Natural Subcategories* | 22 | |
| **Music & Instruments** | **40** | **20.7** |
| Musical Compositions | 14 | |
| Keyboard-Piano | 7 | |
| String-Guitar | 7 | |
| *Other Music Subcategories* | 12 | |
| **Daily Life** | **22** | **11.4** |
| Office-Writing/Typing | 4 | |
| Tools-Electronics | 4 | |
| *Other Daily Life Subcategories* | 14 | |
| **Human Voices** | **21** | **10.9** |
| Adults-Conversation | 6 | |
| *Other Human Voices Subcategories* | 15 | |
| **Transportation** | **12** | **6.2** |
| **Industrial & Urban** | **10** | **5.2** |
| **Special Effects** | **9** | **4.7** |
| **Weapons & Explosions** | **7** | **3.6** |
| **Total Classified** | **191** | **93.2** |
| Unclassified | 14 | 6.8 |
| **Grand Total** | **205** | **100.0** |

Table 4: Detailed Audio Classification Statistics for Set 2

| Category / Subcategory | Count | Percentage (%) |
|---|---|---|
| **Natural Environment** | **106** | **35.9** |
| Weather-Wind | 22 | |
| Animals-Birds | 22 | |
| Weather-Rain | 19 | |
| Animals-Dogs | 9 | |
| *Other Natural Subcategories* | 34 | |
| **Music & Instruments** | **54** | **18.3** |
| Musical Compositions | 16 | |
| Keyboard-Piano | 10 | |
| String-Violin | 8 | |
| *Other Music Subcategories* | 20 | |
| **Daily Life** | **28** | **9.5** |
| Office-Writing/Typing | 6 | |
| Tools-Power Tools | 6 | |
| Communication-Phone | 6 | |
| *Other Daily Life Subcategories* | 10 | |
| **Human Voices** | **16** | **5.4** |
| Adults-Conversation | 9 | |
| *Other Human Voices Subcategories* | 7 | |
| **Transportation** | **10** | **3.4** |
| **Industrial & Urban** | **9** | **3.1** |
| **Weapons & Explosions** | **5** | **1.7** |
| **Special Effects** | **2** | **0.7** |
| **Total Classified** | **230** | **78.0** |
| Unclassified | 65 | 22.0 |
| **Grand Total** | **295** | **100.0** |

Table 5: Combined Audio Classification Statistics for Set 1 & Set 2

| Category / Subcategory | Count | Percentage (%) |
|---|---|---|
| **Natural Environment** | **176** | **36.1** |
| Weather-Wind | 40 | |
| Animals-Birds | 32 | |
| Weather-Rain | 23 | |
| Plants-Vegetation | 15 | |
| *Other Natural Subcategories* | 66 | |
| **Music & Instruments** | **94** | **19.3** |
| Musical Compositions | 30 | |
| Keyboard-Piano | 17 | |
| String-Guitar | 13 | |
| String-Violin | 12 | |
| *Other Music Subcategories* | 22 | |
| **Daily Life** | **50** | **10.2** |
| Office-Writing/Typing | 10 | |
| Tools-Power Tools | 8 | |
| *Other Daily Life Subcategories* | 32 | |
| **Human Voices** | **37** | **7.6** |
| Adults-Conversation | 15 | |
| *Other Human Voices Subcategories* | 22 | |
| **Transportation** | **22** | **4.5** |
| Vehicles-Cars | 19 | |
| **Industrial & Urban** | **19** | **3.9** |
| Fire & Combustion | 14 | |
| **Weapons & Explosions** | **12** | **2.5** |
| **Special Effects** | **11** | **2.3** |
| **Total Classified** | **421** | **84.2** |
| Unclassified | 79 | 15.8 |
| **Grand Total** | **500** | **100.0** |

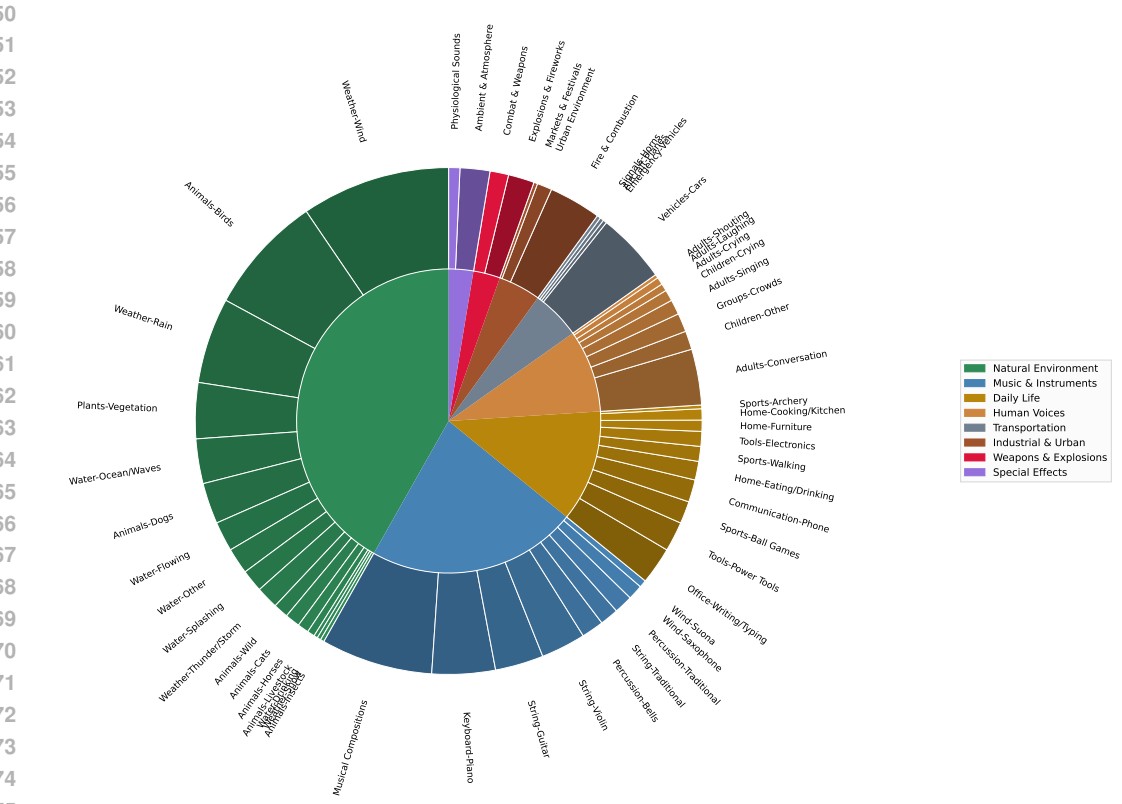

Figure 5: Statistical results of set1 and 2.

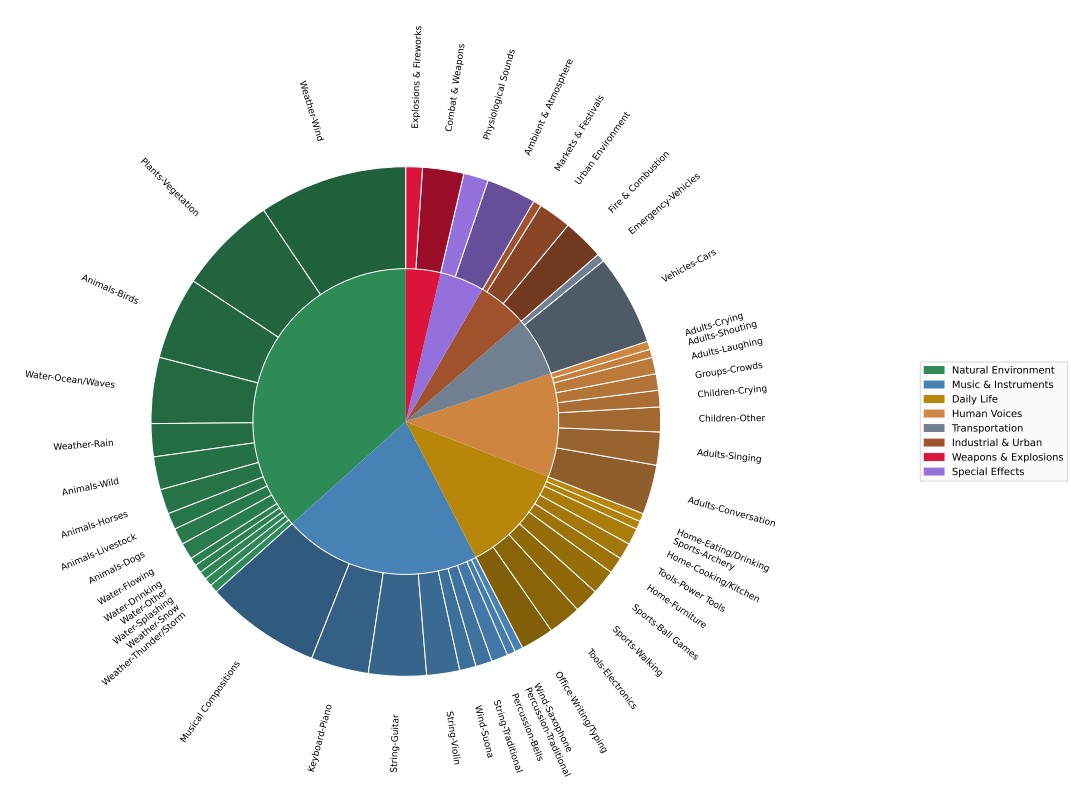

Figure 6: Statistical results of set1.

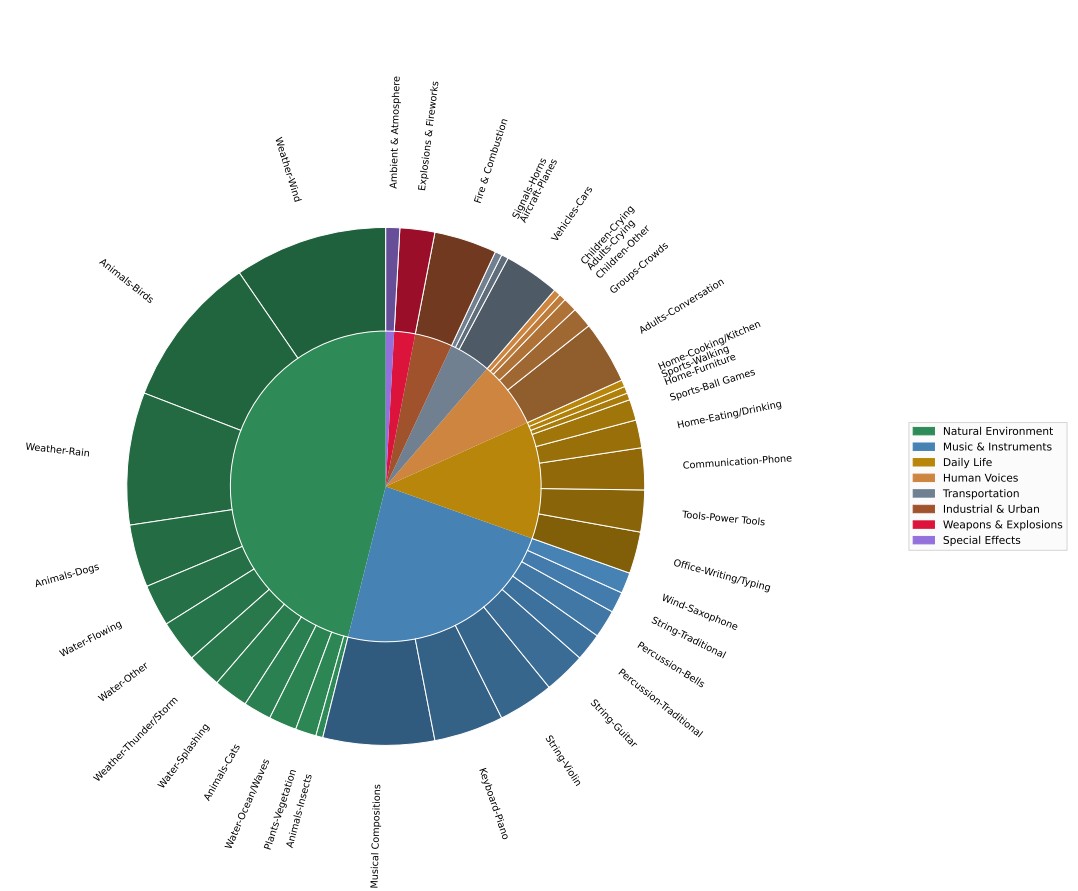

Figure 7: Statistical results of set2.

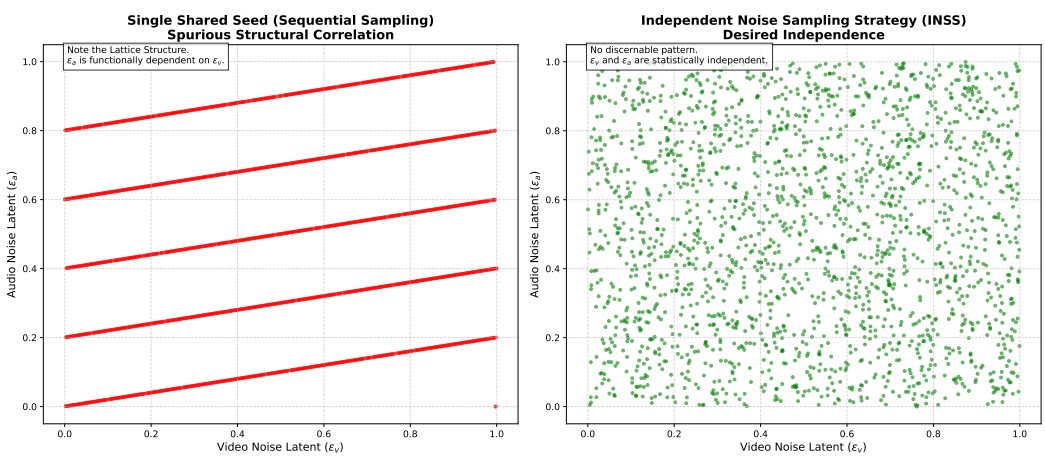

Figure 8: Visualization of effectiveness of INSS in a sample data space.

