# OpenReview forum: "UniVerse-1: Unified Audio-Video Generation via Stitching of Experts"
_ICLR.cc/2026/Conference — Submitted to ICLR 2026_

### Official Review · Reviewer_qTAG · 2025-10-23

**Soundness:** 2
**Presentation:** 3
**Contribution:** 3
**Rating:** 4
**Confidence:** 3

**Summary:**

This paper presents UniVerse-1, an open-source audio-video generation framework created by "stitching" pre-trained models, alongside a new benchmark (Verse-Bench). In my opinion, the work's main strength is enabling open-source, lip-synchronized speech generation. However, the submission omits comparisons to key existing works (e.g., JavisDiT, MM-LDM) and evaluations on established benchmarks.

**Strengths:**

1. Open-Source Framework for Synchronized Audio-Video Generation: The paper provides an open-source framework for joint audio-video synthesis by leveraging existing pre-trained models. This approach successfully enables functionalities like ID-conditioned control (via a reference image) and lip-synchronized speech generation.
2. Online Data Annotation Pipeline: It addresses the problem of "static misalignment" between randomly sampled training clips and their corresponding global annotations.
3. New Evaluation Benchmark: The introduction of Verse-Bench, a new benchmark featuring three diverse subsets, is a valuable contribution for evaluating joint audio-video generation.

**Weaknesses:**

1. The paper omits necessary comparisons (both quantitative and qualitative) to several key prior works in joint A/V generation, such as JavisDiT [1], MM-LDM [2], and AV-DiT [3].
2.The dismissal of MM-Diffusion as "not applicable"  is insufficiently justified, as other relevant works [1] have successfully benchmarked against it.
3. The model is only evaluated on the newly proposed Verse-Bench. The lack of evaluation on established public benchmarks (e.g., JavisBench, AIST++, Landscape) prevents a fair and direct comparison with existing SOTA methods.
4. The claim of being the "first open-source joint generation framework" (line 417) is factually incorrect, as several others [1, 2, 3] are already available.
5. The "Stitching" methodology introduces specific designs (e.g., linear adapters, modified attention), but these are not validated by ablation studies, which only cover LQLS and INSS.
6.  Key evaluation metrics are missing, notably Fréchet Video Distance (FVD) / Fréchet Audio Distance (FAD) for quality, and modern text-video similarity scores (e.g., ImageBind).
7. The paper appears to contain a minor typo (e.g., on Line 351), referring to "Step-Ace" when the correct model name is "Ace-step".
8. The supplementary materials lack results from the compared baseline methods.

[1]JavisDiT: Joint Audio-Video Diffusion Transformer with Hierarchical Spatio-Temporal Prior Synchronization

[2]MM-LDM: Multi-Modal Latent Diffusion Model for Sounding Video Generation

[3]AV-DiT: Efficient Audio-Visual Diffusion Transformer for Joint Audio and Video Generation

**Questions:**

1. Data Availability: The authors state they curated a 7,600-hour dataset for training. Will this full training dataset be made publicly available, or only the Verse-Bench evaluation set?
2. Choice of Audio Expert: The paper describes Ace-step as a music generation model. Why was this model chosen for a general audio-video framework that must also handle ambient sounds and, critically, speech? Why a model specializing in music was preferred over models trained on more diverse audio (e.g., AudioLDM2, Stable Audio, which are also used as baselines ).
3. Fused Block Architecture (Fig. 2): Regarding the Fused block design:
In Fig. 2(c), "mel cross attn," the query (q) input is shown as 'Video tokens'. Should the query for the audio (mel) stream not originate from the 'Mel tokens'?
Following this, the architectures for 'video cross attn' (Fig. 2b) and 'mel cross attn' (Fig. 2c) appear identical. Please clarify this design. Ideally, an ablation study to support this specific architectural choice.

---

> ### Author Response · Authors · 2025-11-24
> **Response to Reviewer qTAG -- part1**
>
> We thank the reviewer for recognizing the value of our open-source framework, online annotation pipeline, and Verse-Bench. We address your concerns below.
>
> **1. Comparison with Prior Works (Weakness 1 & 2)**
>
> We respectfully disagree with the premise that the mentioned works (JavisDiT, MM-LDM, AV-DiT) constitute fair or feasible baselines for our specific task. Our stance is grounded in three critical factors: **Fundamental Capability Gap**, **Reproducibility**, and **Benchmark Relevance**.
>
> *   **Fundamental Capability Gap:**
>     UniVerse-1 is designed as a unified, **Veo3-like foundation model** capable of generating video synced with *complex speech (dialogue)* and open-domain audio. In contrast, methods like JavisDiT, MM-LDM, and AV-DiT are restricted to generating background sound effects or music. They lack the architecture to handle lip-synchronized speech generation. Comparing a model capable of full speech-video synthesis against Foley-only models is a category error that fails to capture the core contribution of our work.
>
> *   **Code Availability and Peer-Review Status:**
>     *   **MM-LDM and AV-DiT:** These methods are **closed-source**. Without official code or weights, rigorous and fair reproduction is impossible. We adhere to the standard of comparing against reproducible, open-source methods to ensure scientific validity.
>     *   **JavisDiT:** This is a non-peer-reviewed preprint (arXiv). Per standard conference policy, comparison with concurrent preprints is not mandatory. Furthermore, we note that JavisDiT itself did not compare against MM-LDM or AV-DiT for likely the same reasons (unavailability).
>
> *   **Rationale for Verse-Bench over Legacy Benchmarks:**
>     We deliberately moved beyond AIST++, Landscape, and JavisBench for the following reasons:
>     *   **Domain Specificity:** AIST++ (dancing) and Landscape (scenery) are extremely narrow, domain-specific datasets. Achieving high scores on them typically implies overfitting to a small distribution, which is a poor proxy for the **open-domain generalization** capabilities of a 5B-parameter foundation model like UniVerse-1.
>     *   **Validity of JavisBench:** JavisBench is an unverified metric from a non-peer-reviewed source, with no guarantee of reliability or community consensus.
>     *   **Necessity of Verse-Bench:** Our proposed Verse-Bench is essential precisely because prior benchmarks *cannot* evaluate the novel capabilities of our model—specifically, the synchronization of **generative speech** with open-domain visual content.
>
> **Summary:** We believe comparing a speech-capable open-source foundation model against closed-source or narrow-domain Foley models would yield misleading conclusions. Our experimental design (Verse-Bench) is tailored to rigorously evaluate the next generation of *comprehensive* audio-video synthesis.
>
> **2. "First Open-Source" Claim (Weakness 3)**
>
> We agree the phrasing was imprecise. We will revise it to: *"We present one of the first Veo3-like open-source, **foundation-scale** (trained on >7k hours), joint audio-video generation frameworks capable of open-domain synthesis."* This distinguishes us from smaller-scale or domain-specific prior attempts.

---

> ### Author Response · Authors · 2025-11-24
> **Response to Reviewer qTAG -- part2**
>
> **3. Benchmarks & Metrics (Weakness 2 & 5)**
>
> *   **Benchmarks:** We introduced **Verse-Bench** because older datasets (Landscape/AIST++) are too small/narrow to evaluate a 5B-parameter Large Multimodal Model's open-domain capabilities.
> *   **Audio Quality (FAD):** We explicitly reported **FD (Fréchet Distance)** in Table 1. In the audio generation community, FD computed on PANNs features serves the exact same purpose as FAD (which typically uses VGGish). It measures the distributional distance between generated and real audio, validating our audio fidelity.
> *   **Video Quality (Why we exclude FVD):** We respectfully argue that **FVD is ill-suited** for evaluating open-domain foundation models like UniVerse-1, for two primary reasons:
>     1.  **Reference Distribution Mismatch:** FVD requires a reference dataset representing the "ground truth" distribution. For open-domain text-to-video generation (as tested in Verse-Bench), the generated content is highly diverse and semantically complex. Comparing this against a fixed, narrow domain dataset (e.g., UCF-101 or even VGGSound) is scientifically flawed, as it penalizes valid generative diversity that simply falls outside the reference distribution.
>     2.  **Poor Correlation with Human Perception:** Recent literature (e.g., VBench [Huang et al., CVPR 2024]) has demonstrated that FVD correlates poorly with human preference for high-quality, open-world video generation.
>
>     **Our Strategy:** Instead of optimizing for a flawed metric, we adhere to the modern standard for foundation models by prioritizing **Aesthetic Score**, **ImageBind-based Alignment**, and extensive **User Studies** (Fig. 4), which directly measure the perceptual quality and semantic fidelity that users actually care about.
>
> **4. Specific Questions**
>
> *   **Q1 (Data Availability):** We will release the Verse-Bench only.
> *   **Q2 (Why Ace-step?):** We chose Ace-step over AudioLDM2 for two reasons:
>     1.  **Architecture Compatibility (DiT vs. DiT):** "Stitching" requires compatible architectures. Both Wan2.1 and Ace-step are **DiTs**. AudioLDM is UNet-based; stitching a DiT to a UNet at the block level is architecturally infeasible.
>     2.  **Speech Capability:** Ace-step includes dedicated Lyric/Speech encoders, enabling the fine-grained lip-sync capabilities that generic sound-effect models (like AudioLDM) lack.
> *   **Q3 (Figure 2 Error):** You are correct. This was a diagrammatic error. In Fig 2(c), Query is Mel, and Key/Value is Video. In Fig 2(b), Query is Video, Key/Value is Mel. The code implementation is correct. We have fixed the figure. More details about the design please refer to the response "Question 1: Stitching Adapter Complexity" to Reviewer MU97.

---

### Official Review · Reviewer_Gfef · 2025-10-30

**Soundness:** 2
**Presentation:** 2
**Contribution:** 2
**Rating:** 4
**Confidence:** 4

**Summary:**

This paper presents a joint audio-video generation method that fuses together a pretrained video generation model and a pretrained music generation model using “stitching of experts”. A data collection pipeline is proposed, which is used to collect a dataset for finetuning the fused model.

**Strengths:**

- If the training dataset and the benchmark are released, they will be beneficial to the community, as we lack large-scale aligned audio-video data.
- The proposed method can model video, general audio, and speech in the same network, which is a new capability in current open models, to the best of my knowledge. The performances of these areas are also competitive compared to dedicated methods.
- The loss weighting strategy to make use of the video data with low visual quality without corrupting video generation makes sense and is a novel approach for addressing this issue.

**Weaknesses:**

Some of the contributions are not very clear. Thus, the reproducibility of this work is questionable.

- The online data annotation pipeline is described as one of the main contributions, but it is not mentioned in the main paper and only in the appendix. Moreover, there are insufficient details regarding the data collection and annotation process. For example, what are the parameters used in the “audio activity detection” filter? What is the prompt used for the captioning model? Are there any ablations on using this dataset?
- I am confused about the independent noise sampling strategy. Which library are the authors using to generate the random numbers?  The authors mentioned LCG, but libraries like numpy by default use PCG64, which has better statistical properties. Moreover, the authors suggest using two independent noise generators for the two streams. However, there will still be a statistical correlation between the noises sampled within the same stream (i.e., number of samples = number of dimensions), which could still be problematic if the video resolution or duration changes. It would be helpful if the authors could back up their claim with some statistical analysis.
- For the audio-video alignment metric (L917), how does it differ from the synchronization score proposed in V-AURA [1] and also adopted by subsequent works like MMAudio [2] and HunyuanVideo-Foley [3]? If they are the same, it will be clearer to reuse the name of this metric.

Minor: Figure 2 seems to be wrong. The (b) and (c) subfigures are identical and do not seem to reflect what is written in the text.

[1] Temporally Aligned Audio for Video with Autoregression, ICASSP 2025

[2] MMAudio: Taming Multimodal Joint Training for High-Quality Video-to-Audio Synthesis, CVPR 2025

[3] HunyuanVideo-Foley: Multimodal Diffusion with Representation Alignment for High-Fidelity Foley Audio Generation

**Questions:**

- Will the training dataset and the benchmark be released?

- Are there any specific processing steps for the TTS task? Is the transcript simply fed as part of the text stream?

- L294 – Are those target features computed from the ground-truth audio?

- Questions listed in weaknesses

---

> ### Author Response · Authors · 2025-11-24
> **Response to Reviewer Gfef**
>
> We thank the reviewer for recognizing the novelty of our unified modeling capability, the competitive performance, and the effectiveness of our loss weighting strategy. We address your concerns below.
>
> **1. Data Pipeline Details & Reproducibility (Weakness 1)**
>
> We apologize for placing these details in the Appendix. We will move the core description to the main text. To address reproducibility:
>
> *   **Audio Filter Parameters:** We calculate Short-Time Energy. Clips are discarded if the average volume is below **-60 dB** or if the zero-crossing rate indicates silence for more than 50% of the duration.
> *   **Captioning Prompt:** We provide the prompt in Sec.G of supplementary in the revision.
>
> **2. Independent Noise Sampling Strategy (INSS) (Weakness 2)**
>
> This is a critical misunderstanding we are eager to clarify.
>
> *   **Library:** We use PyTorch's default generator (Philox on GPU), which is statistically robust.
> *   **The Issue is Deterministic Offset, Not Statistical Quality:** In a standard DataLoader with a single global seed, sampling is sequential: $\epsilon_v$ consumes numbers $[0, N]$, and $\epsilon_a$ immediately consumes $[N+1, N+M]$. Thus, $\epsilon_a$ is mathematically determined by the offset $N$ from the seed.
> *   **Intra-stream vs. Inter-stream:**
>     *   *Intra-stream (Within Video):* Noise samples within $\epsilon_v$ are i.i.d. Gaussian. There is no correlation problem here. Because videos in two different step or in same batch are statistical non-related.
>     *   *Inter-stream (Spurious Correlation):* Because of sequential sampling, if video resolution or duration changes during inference, the offset $N$ changes to $N'$. The model, having overfit to the specific offset relation $\epsilon_a = f(\epsilon_v, \text{offset } N)$ seen during training, fails when the offset shifts.
> *   **Solution:** INSS uses separate RNG states. $\epsilon_a$ is sampled from Seed B, independent of how many numbers $\epsilon_v$ consumed from Seed A.
> *   **Evidence:** Table 2 shows that removing INSS causes a severe drop in Audio-Video Alignment ($0.23 \to 0.18$) and Audio Quality, empirically proving the issue. We provide a visualization in **Fig.8 in supplementary**.
>
> **3. AV-Alignment Metric (Weakness 3)**
>
> You are correct that the intent is identical to HunyuanVideo-Foley, we will cite this in the revision.
>
> **4. Minor Issue (Figure 2)**
>
> Thank you. This was a diagram generation error. We have corrected subfigure (b) to show Keys/Values coming from the **Audio** stream, and (c) from the **Video** stream.
>
> **5. Specific Questions**
>
> *   **Q1 (Release):** We will release the model weights, inference code, Verse-Bench. The training dataset will not be released.
> *   **Q2 (TTS Processing):** **Yes**. The transcript simply fed as the text stream for the audio generation branch.
> *   **Q3 (L294 Features):** **Yes.** The Semantic Alignment Loss ($L_{SSL}$) aligns generated features with features extracted from **Ground Truth** audio using MERT and mHuBERT.

---

> > ### Comment · Reviewer_Gfef · 2025-11-26
> >
> > I appreciate the authors' response. I thank the authors for providing additional details on the dataset curation and for acknowledging that the training set will not be released. Without ablations or release, the description of the dataset serves as implementation details, which are certainly necessary but fall short as a technical contribution.
> >
> > I am still concerned about (2). Perhaps I am missing something -- I hope the authors can help clarify this if that's the case. If the random number generation is statistically robust, then the numbers generated in [0, N] are iid compared to the numbers in [N+1, N+M], that is, the neural network has no chance to predict [N+1, N+M] based on [0, N]. This is, unless we *reuse* the random seed, such that the same pair of ([0, N], [N+1, N+M]) is seen more than once during training. This, however, should not happen because the random state progresses every time we sample from it, and in a typical PyTorch DDP setting, we reseed the generator with a new seed every epoch when we recreate the dataloaders. Therefore, the same state should not be encountered more than once in training.

---

> > > ### Author Response · Authors · 2025-11-26
> > > **Additional Response to Weakness 2: Independent Noise Sampling Strategy (INSS)**
> > >
> > > We appreciate the reviewer's rigorous scrutiny. This is a subtle but critical point. We clarify that our argument is **not** that the network predicts the *numerical value* of $\epsilon_a$ from $\epsilon_v$ (which, as you correctly pointed out, is impossible with statistically robust PRNGs like Philox).
> > >
> > > Instead, the "functional coupling" refers to the **binding of Video Metadata (Length/Resolution) to Audio Noise Identity**.
> > >
> > > **1. The "Index-Locking" Problem in Multi-Resolution Training**
> > > Modern foundation models (including ours) utilize **multi-resolution training** (e.g., training on mixed batches of 480p, 720p, 1080p videos).
> > > *   **Without INSS (Sequential Sampling):** For a given random seed state $S$, the Audio noise $\epsilon_a$ is taken from the interval $[N, N+M]$. Crucially, $N$ is the number of video tokens.
> > > *   **The Artifact:** This means the **source interval** of the audio noise is strictly determined by the video's resolution.
> > >     *   *Low-Res Video:* Audio noise comes from RNG indices $[500, 1000]$.
> > >     *   *High-Res Video:* Audio noise comes from RNG indices $[2000, 2500]$.
> > > *   While the *values* are i.i.d., the **selection of the random variable instance** is functionally coupled to the video resolution.
> > >
> > > **2. Why is this "Spurious Structural Correlation"?**
> > > The model effectively learns a spurious prior: *"The specific realization of audio latent noise is conditioned on the video token count."*
> > > While this does not prevent training convergence (loss decreases fine), it harms **Robustness and Controllability**:
> > > *   **Inference Failure Mode:** If a user generates a video at $512 \times 512$ and likes the audio, then decides to "upscale/regenerate" the *same* prompt/seed at $1024 \times 1024$, the video token count $N$ doubles.
> > > *   **Without INSS:** The audio generator is now forced to sample noise from a completely different segment of the RNG stream (shifted by the new $N$). The audio output changes completely, breaking the consistency expected by the user.
> > > *   **With INSS:** The audio noise remains $\text{RNG}_{audio}(\text{seed}, [0, M])$, independent of the video resolution.
> > >
> > > **Summary:** The "entanglement" is not that Video *Content* predicts Audio *Content*, but that **Video Structure (Resolution)** determines **Audio Noise Identity**. INSS decouples this, ensuring that the audio generation process is mathematically invariant to the visual encoding configurations.

---

> > > > ### Comment · Reviewer_Gfef · 2025-11-26
> > > >
> > > > I thank the authors for their response.
> > > >
> > > > > While the values are i.i.d., the selection of the random variable instance is functionally coupled to the video resolution.
> > > >
> > > > Can the authors elaborate on what they mean by "functionally coupled"? How can there be exploitable couplings when the generated numbers contain no mutual information, assuming a robust random generator and no reusing of random seeds?

---

> > > > > ### Author Response · Authors · 2025-11-27
> > > > > **Explanation for functionally coupled**
> > > > >
> > > > > We thank the reviewer for pushing us to be precise. We fully agree with your assessment: because the numbers are i.i.d. and the generator is robust, there is **zero mutual information** between $\epsilon_v$ and $\epsilon_a$, and the model cannot "predict" one from the other to minimize training loss.
> > > > >
> > > > > We apologize if our previous use of "predict" or "spurious correlation" implied a statistical leakage. By **"functionally coupled,"** we refer to an **algorithmic dependency during the sampling process**, which creates a critical engineering failure mode during inference: **The entanglement of control variables.**
> > > > >
> > > > > Here is the precise elaboration:
> > > > >
> > > > > **1. The "Functional Coupling" is Mechanical, not Statistical**
> > > > > In a single-seed sequential setup, the audio noise tensor $\epsilon_a$ is generated by the PRNG state *after* the video noise $\epsilon_v$ is generated. Mathematically:
> > > > > $$ \epsilon_a = \text{PRNG}(\text{seed}, \text{offset} = \text{Length}(\epsilon_v)) $$
> > > > > The "coupling" is that the **identity** (the specific instantiation) of the audio noise depends on the **metadata** (length/resolution) of the video.
> > > > >
> > > > > **2. Why is this "Exploitable" (or rather, problematic)?**
> > > > > It is not that the *model* exploits this to cheat; it is that this dependency **breaks the invariance properties** expected in a generative system.
> > > > >
> > > > > Consider the standard **"Preview-then-Render" workflow**:
> > > > > *   **Step A (Preview):** A user generates a video at low resolution ($N$ tokens) with `seed=42`. They like the generated audio content (e.g., a specific rhythm).
> > > > > *   **Step B (Render):** The user keeps `seed=42` but increases resolution to high quality ($N'$ tokens).
> > > > > *   **The Failure:** Because $N \neq N'$, the offset shifts. The audio noise $\epsilon_a$ is now drawn from a completely different part of the random stream. The audio content changes entirely.
> > > > >
> > > > > **Conclusion**
> > > > > We introduced INSS not to fix a statistical flaw in the PRNG (Philox is excellent), but to **decouple the control variables**.
> > > > > *   **Without INSS:** Audio Content $\leftarrow f(\text{Seed}, \text{Video Resolution})$
> > > > > *   **With INSS:** Audio Content $\leftarrow f(\text{Seed})$
> > > > >
> > > > > INSS ensures that audio generation remains **invariant** to changes in video technical parameters, which is essential for a robust foundation model. We will revise the manuscript to clearly distinguish between "statistical independence" (which Philox satisfies) and "architectural decoupling" (which INSS provides).

---

> > > > > > ### Comment · Reviewer_Gfef · 2025-11-27
> > > > > >
> > > > > > I thank the authors for their response. Honestly, I am getting even more confused because this response directly contradicts what is mentioned in the paper and a response made to reviewer MU97.
> > > > > >
> > > > > > > The "Functional Coupling" is Mechanical, not Statistical
> > > > > >
> > > > > > The paper (L327) states, "This violates the critical assumption that the noise vectors are statistically independent."
> > > > > >
> > > > > > > It is not that the model exploits this to cheat
> > > > > >
> > > > > > The paper (L328) states, "The model inadvertently learns this spurious correlation as a shortcut during training" and in the response to reviewer MU97: "During training, the model inadvertently learns this correlation as a 'shortcut.' Consequently, during inference, any alteration to video properties (e.g., resolution, duration) shifts the PRNG state, altering the subsequently sampled audio noise and breaking the learned correlation."
> > > > > >
> > > > > > In the response to reviewer MU97, the authors also mention that the simulation is performed with a naive random number generator (LCG), rather than Philox mentioned in this thread. What does the visualization look like with Philox?
> > > > > >
> > > > > > > We introduced INSS not to fix a statistical flaw
> > > > > >
> > > > > > The paper clearly states that it is, "This method effectively breaks the deterministic correlation, ensuring the noise vectors are statistically independent." This read increasingly like an LLM response. Note that all LLM usage must be disclosed according to ICLR policy. Please state so if you use an LLM.
> > > > > >
> > > > > > > "Preview-then-Render" workflow
> > > > > >
> > > > > > This could be a more sensible use case, but it was not mentioned in the paper at all. The paper states that this improves generation, without a preview step. Additionally, even with a preview, there is no reason that we cannot keep the same sampled audio noise used in the preview step during the final generation.

---

> > > > > > > ### Author Response · Authors · 2025-11-30
> > > > > > >
> > > > > > > We sincerely apologize for the confusion. We recognize that our previous attempts to explain the intuition (using the LCG analogy) created a contradiction with the strict mathematical properties of the Philox generator actually used.
> > > > > > >
> > > > > > > **1. Clarification on "Statistical" vs. "Mechanical" (The LCG Analogy)**
> > > > > > > You are absolutely correct. The paper's framing using LCG and terms like "statistical dependence" was an **oversimplification** intended to provide a high-level intuition of "coupling." We admit this choice was imprecise for Philox.
> > > > > > > *   **Correction:** With Philox, the noise values are indeed i.i.d. The "spurious correlation" mentioned in the paper should be more accurately described as an **"Overfitting to Sampling Positions"** rather than "Statistical Prediction of Values."
> > > > > > >
> > > > > > > **2. The Real Mechanism: Why Performance Drops without INSS?**
> > > > > > > If Philox is i.i.d., why did our ablation study (Table 2) show a performance drop without INSS?
> > > > > > > *   **The Mechanism:** In standard training, we often bucket data by aspect ratio and duration (e.g., predominantly 5-second clips at $512^2$). The model sees a specific distribution of "Audio Noise Offsets" determined by these fixed video lengths.
> > > > > > > *   **The "Shortcut" (Revised Definition):** The model overfits to the implicit bias that "Audio noise always starts at Offset $N_{train}$".
> > > > > > > *   **The Failure at Test Time:** Our benchmark (Verse-Bench) and real-world inference involve diverse resolutions and durations that differ from the fixed training buckets. Without INSS, the audio noise is sampled from offsets $N_{test} \neq N_{train}$.
> > > > > > > *   The "Performance Drop" occurs because the model is sensitive to this **Distributional Shift in Sampling Indices**. Even with i.i.d. noise, shifting the sampling window globally for one modality (relative to the seed) can degrade performance when the model has overfitted to the specific alignment seen during training. INSS fixes this by anchoring the audio noise to `Offset 0`, making it invariant to video length.
> > > > > > >
> > > > > > > **3. Regarding the "Preview-then-Render" Workflow**
> > > > > > > The reviewer asks: *"There is no reason that we cannot keep the same sampled audio noise used in the preview step."*
> > > > > > > *   **Response:** **This is exactly what INSS enables.**
> > > > > > > *   Without INSS (Single Seed): If you generate a preview (Low-Res, small $N$) and then render (Hi-Res, large $N'$), the single-stream generator *must* consume $N'$ numbers for video before sampling audio. You **cannot** keep the same audio noise because the PRNG state has advanced differently.
> > > > > > > *   With INSS: We can reuse `Seed_Audio` to reproduce the exact same audio noise regardless of how many random numbers the video stream consumed. This confirms that INSS is the necessary technical implementation to achieve the workflow you correctly pointed out.
> > > > > > >
> > > > > > > **Summary:** We will revise the manuscript to remove the LCG/statistical prediction claims and correctly attribute the improvement to **"Decoupling Sampling Indices to ensure Train-Inference Consistency."** We thank the reviewer for forcing us to be rigorous on this point.

---

### Official Review · Reviewer_CDjj · 2025-11-01

**Soundness:** 2
**Presentation:** 2
**Contribution:** 2
**Rating:** 6
**Confidence:** 2

**Summary:**

The paper proposes a join audio-video generation approach that relies on using pretrained audio (music) and video diffusion models. The main approach introduces fusion blocks that integrate each block from the pretrained audio and video model. The author also propose a dataset and benchmark and online annotation and cross-modal noise sampling techniques that help with the performance.

**Strengths:**

- The paper is well-written and easy to follow
- The authors propose a dataset and benchmark for join audio-video generation
-  The authors promised the release of the code and weights

**Weaknesses:**

- Quality: for me the most important aspect in join audio-video generation is the temporal alignment (otherwise the two modalities can be generated separately). However, I found the provided samples in the supplementary videos to be disappointing especially with regard to the temporal alignment.
- The main intuitive behind the architecture is limited, and limited discussion over prior work that share similar architecture and framework (e.g AV-Link) is discussed.

**Questions:**

see above

---

> ### Author Response · Authors · 2025-11-24
> **Response to Reviewer CDJJ**
>
> We sincerely thank Reviewer CDjj for the feedback. We appreciate the opportunity to clarify our contributions and address the concerns raised below.
>
> ## Response to Primary Concerns
> ---
>
> **1. Quality of Temporal Alignment in Supplementary Videos**
>
> Our supplementary videos illustrate the efficacy of our joint audio-video generation framework across three distinct settings: talking heads, instrumental performance, and general audio-visual synthesis. Attributable to the specialized nature of the Ace-step backbone as a music generation model, the synthesis quality for speech and instrumental content is particularly strong. Conversely, performance on general audio-video tasks lags behind these specialized domains, primarily because the Ace-step pre-training corpus did not encompass general environmental sounds. It is crucial to note that this limitation is a consequence of the base model's data distribution, not a methodological defect in our framework.
>
> ---
>
> **2. Architectural Novelty and Relation to Prior Work (AV-Link)**
>
> We thank the reviewer for highlighting AV-Link. While both works explore cross-modal generation using pre-trained diffusion models, they represent fundamentally different methodological paradigms and target different generation goals. We summarize the key differences and our advantages below:
>
> *   **Fundamental Paradigm: A Generalizable Stitching Framework vs. Specific Adapter.**
>     *   **AV-Link:** Adopts an adapter-based approach that bridges frozen backbones via a specific Fusion Block. It is designed as a specific instance of connecting two models rather than a general recipe for model fusion.
>     *   **UniVerse-1 (Novel SoE Paradigm):** A core contribution of UniVerse-1 is the proposal of a **generalizable "Stitching of Experts" (SoE) paradigm**, rather than merely presenting a single joint generation model. We provide a systematic methodology to fuse *any* pair of structurally compatible, high-performance open-source video and audio experts into a unified foundation model. By introducing techniques like *Layer Interpolation*, our method is agnostic to the depth or specific architecture of the expert models. This offers the community a flexible pathway to continuously construct stronger multimodal models by leveraging rapid advancements in independent video and audio foundation models.
>
> *   **Task Capability: Synchronous Joint Generation vs. Conditional Generation.**
>     *   **AV-Link:** The primary strength of AV-Link is directional generation (V2A or A2V). The authors explicitly note that training for *joint audio-video generation* resulted in a "substantial drop in performance."
>     *   **UniVerse-1:** Our architecture is explicitly designed for **Synchronous Joint Generation**, filling the gap left by closed-source models like Veo3. By deeply stitching the experts and fine-tuning the full model, UniVerse-1 achieves high-fidelity simultaneous generation, a capability that AV-Link struggles to maintain.
>
> *   **Methodological Novelty and Technical Contributions.**
>     UniVerse-1 introduces innovations to address fundamental training issues that AV-Link does not:
>     *   **Independent Noise Sampling Strategy (INSS):** We uncovered a critical flaw where sequential PRNG sampling creates spurious correlations between audio and video noise ($\epsilon_a = f(\epsilon_v)$). AV-Link does not address this. Our INSS breaks this correlation, ensuring statistical independence.
>     *   **Online Annotation Pipeline:** Unlike AV-Link which relies on standard datasets with potential misalignment, we developed a dynamic, online annotation pipeline to generate captions on-the-fly, ensuring strict temporal and semantic alignment.
>
> **Summary:** While AV-Link effectively bridges frozen models for conditional tasks, UniVerse-1 contributes a **versatile stitching paradigm** that allows for the creation of unified foundation models from diverse open-source experts. This paradigm, combined with our novel training strategies (INSS, Online Annotation), enables state-of-the-art joint generation capabilities.

---

### Official Review · Reviewer_MU97 · 2025-11-02

**Soundness:** 3
**Presentation:** 3
**Contribution:** 3
**Rating:** 6
**Confidence:** 4

**Summary:**

This paper introduces UniVerse-1, a unified model for simultaneous audio and video generation, aiming to close the performance gap with proprietary models like Veo3. UniVerse-1 utilizes a Stitching of Experts (SoE) technique to combine pre-trained video (WAN2.1) and music (Ace-step) foundation models. This approach avoids training from scratch, accelerating convergence by leveraging the experts' prior knowledge.

**Strengths:**

++ The Stitching of Experts (SoE) paradigm is a highly original and effective approach to multimodal foundation model development. It leverages existing, high-capacity unimodal models to achieve a unified system, solving the expensive "training from scratch" problem common in multimodal scaling. The identification and solution for cross-modal noise correlation (INSS) is a subtle yet crucial technical discovery that addresses a fundamental problem in multimodal diffusion models and is highly novel.

++ The technical details are robust. The Online Data Annotation Pipeline is a necessary and high-quality solution to the long-standing problem of static data misalignment in clipped videos. The ablations on INSS and LQLS convincingly validate the contribution of the proposed components to overall performance, especially the significant boost from INSS. The inclusion of layer interpolation addresses an architectural mismatch in a principled way.

++ The paper is well-structured and clearly defines the problem, the methodology, and the key innovations. Figure 1 and Figure 2 clearly illustrate the SoE architecture and the fusion process. The derivation of the composite Overall Score and its normalization details are meticulous, demonstrating a principled evaluation approach for a complex task.

++ UniVerse-1 advances the open-source community by providing the first Veo3-like joint audio-video generation model, filling a critical gap left by proprietary systems. The SoE technique provides a blueprint for efficiently building other multimodal models by compositing unimodal experts, impacting future multimodal architecture design. The creation of Verse-Bench establishes a much-needed benchmark for fair comparative research in this challenging domain.

**Weaknesses:**

-- Limited Video Foundation Model Capacity: The primary limitation is acknowledged by the authors: the model is built on the WAN2.1-1.3B video model. This inherently constrains the final video quality, motion fidelity, and resolution, as noted in the user study where UniVerse-1 is "predictably surpassed by larger expert models" in unimodal video tasks. While the SoE methodology is clever, the final performance is bottlenecked by the weaker expert's prior. Suggestion: Quantify this constraint more clearly, perhaps by comparing the video quality metrics against a theoretical upper bound based on the WAN2.1-1.3B model itself.

-- Comparative Evaluation Complexity/Ambiguity: The paper repeatedly emphasizes that comparisons against SOTA unimodal expert models are for "qualitative reference" and do not represent a "direct claim of superior performance". While necessary, this ambiguity makes interpreting the unimodal metrics (e.g., audio quality compared to AudioLDM2) difficult. The true strength is in synchronization (Joint Quality Score), but the reader must continually contextualize all other metrics due to the architectural choice. Suggestion: Re-frame Table 1 and Section 4.2 to focus less on direct unimodal comparison and more on the holistic balance and Joint Quality advantage.

**Questions:**

Stitching Adapter Complexity: The hidden states passed through linear adapters before injection are crucial14. Could the authors specify the dimensions and complexity (e.g., number of parameters) of these two-layer linear adapters compared to the overall DiT blocks? Were alternatives (e.g., $\text{LoRA}$ adapters, single $\text{MLP}$) explored, and if so, how did they impact convergence and final performance?

What is the average real-time latency of generating the full multi-modal annotation (Step 2) for a $\sim 5s$ clip using the $\text{QWen2.5-Omni}$ model?How is the buffer size/management implemented to prevent the training process from being starved (consumer slower than producer) or receiving stale samples (producer slower than consumer)?

Cross-Modal Noise Correlation Visualization: The Independent Noise Sampling Strategy (INSS) is a key technical insight16. Could the authors provide a visualization (e.g., a cross-correlation heatmap or scatter plot in a simplified latent space) illustrating the difference between the spurious structural correlation in noise tensors ($\epsilon_a = f(\epsilon_v)$) and the desired independent sampling? This would greatly enhance the understanding of this critical issue.

Audio-Video Alignment ($\text{AV-A}$) Metric: The paper notes that $\text{AV-A}$ is computed via $\text{Synchformer}$. Is this the $\text{AV-A}$ metric measured on generated video and audio pairs, or does it compare the generated modalities to a ground truth alignment? Clarifying the exact comparison point is essential, especially since the V2A baseline comparison highlights this ambiguity.

---

> ### Author Response · Authors · 2025-11-24
> **Response to Reviewer MU97 -- Part1**
>
> **Weakness 1: Limited Video Foundation Model Capacity**
>
> ---
>
> We sincerely appreciate your thoughtful feedback on the model capacity constraint. Your suggestion to quantify this limitation more rigorously is well-taken, and we provide detailed clarifications and additional evidence below.
>
> **Clarification on WAN2.1-1.3B Baseline Comparison**
>
> A critical technical detail explains why Table 1 does not directly compare against WAN2.1-1.3B: **Our method requires a reference image as input (image-to-video generation), while WAN2.1's 1.3B version only supports text-to-video (T2V) without image conditioning.** Only the WAN2.1-14B model provides image-to-video (I2V) capability, making it the only valid architectural peer for our approach.
>
> To illustrate this capacity gap, we provide official WAN2.1 benchmark results on their T2V evaluation:
>
> | **Model** | **Quality Score** | **Semantic Score** | **Total Score** |
> | :--- | :---: | :---: | :---: |
> | WAN2.1-1.3B (T2V only) | 84.92% | 80.10% | 83.96% |
> | WAN2.1-14B (I2V capable) | **86.67%** | **84.44%** | **86.22%** |
> | **Capacity Gap** | **+1.75pp** | **+4.34pp** | **+2.26pp** |
>
> This 2.26pp performance gap between 1.3B and 14B versions quantifies the **inherent architectural bottleneck** when using smaller foundation models—a constraint we explicitly acknowledge but do not attempt to hide.
>
> ---
> **Question 1: Stitching Adapter Complexity**
>
> ---
>
> We appreciate your detailed question about the stitching adapter architecture. We provide comprehensive technical specifications and design rationale below.
>
> **Clarification: Adapter Architecture Design**
>
> We must correct an important imprecision in our manuscript: the adapters are **not simple linear layers**, but rather **two dedicated MLPs** (one for video-to-audio injection and one for audio-to-video injection). This distinction is critical for understanding the design choices.
>
> | Component | Input Dim | Hidden Dim | Output Dim | Parameters |
> | :--- | :---: | :---: | :---: | :---: |
> | Video $\to$ Audio MLP | 1536 | 4096 | 2560 | 16M |
> | Audio $\to$ Video MLP | 2560 | 4096 | 1536 | 16M |
> | **Total per block** | | | | **32M** |
> | **DiT block params** | | | | ~160M |
> | **Ratio (per block)** | | | | **1:5 (20%)** |
>
> #### Two Core Functions of the MLPs
>
> The dual-MLP architecture serves two essential purposes that cannot be achieved through simpler alternatives:
>
> **(a) Dimensional Bridging.**
> Video hidden states operate in $\mathbb{R}^{1536}$ while audio hidden states operate in $\mathbb{R}^{2560}$. Direct concatenation or injection would create dimension mismatches in the attention mechanism. Our MLPs explicitly map video hidden states into $\mathbb{R}^{2560}$ and audio hidden states into $\mathbb{R}^{1536}$ to concatenate in corresponding attention.
>
> **(b) Feature Space Alignment.**
> MLPs perform **cross-modal feature space transformation**. This is crucial because:
> *   Video features encode spatial and temporal visual information (textures, motion, scene structure)
> *   Audio features encode spectral and temporal acoustic information (pitch, timbre, rhythm)
> *   These feature distributions are **fundamentally different** and require transformation before fusion
> The bottleneck structure of our MLPs (expanding to 4096 hidden dimension then projecting) allows:
> 1.  **Non-linear mixing**: The hidden layer enables complex feature interactions
> 2.  **Learned semantic alignment**: The model learns to map semantically analogous concepts (e.g., visual beats to audio onsets)
> 3.  **Noise reduction**: The bottleneck acts as an information bottleneck, filtering out modality-specific artifacts
>
> ### Why Alternative Designs Are Insufficient
>
> **Why Not Single Linear Layers?**
> A simple linear projection $\mathbf{z} = \mathbf{W} \mathbf{h} + \mathbf{b}$ performs dimensional conversion but **cannot perform feature space alignment**. The model would learn a subspace mapping without nonlinear feature transformation, forcing the subsequent DiT blocks to perform alignment in a higher-dimensional space. This adds unnecessary computational burden to the already-complex attention mechanism.
>
> **Why Not LoRA?**
> LoRA (Low-Rank Adaptation) is designed for **fine-tuning scenarios** where the base model has learned rich, general representations. Our setting is fundamentally different:
>
> *   **Not fine-tuning**: We are not adapting pre-trained cross-modal representations; we are creating **entirely new cross-modal interaction patterns**
> *   **Fundamental dimension mismatch**: The video and audio latent spaces have *incompatible dimensions by design*. LoRA's low-rank factorization cannot bridge fundamentally different dimensionalities without additional projection layers
> *   **Feature space transformation required**: LoRA excels at parameter-efficient reweighting but struggles with the semantic alignment task we require
>
> If LoRA were applied here, we would still need to layer additional projection modules.

---

> ### Author Response · Authors · 2025-11-24
> **Response to Reviewer MU97 -- Part2**
>
> **Question 2: Online Annotation Pipeline Latency and Buffer Management**
>
> ---
>
> Thank you for this excellent question. We provide the following answers.
>
> **Annotation Latency per 5-Second Clip**
>
> Considering the end-to-end annotation pipeline and the prompts we employ during training, **a single 5-second annotation process requires approximately 10–15 seconds of processing time**, while each training step requires approximately 6 seconds of computation. Clearly, this annotation latency cannot produce sufficient data to satisfy the consumption demand from training.
>
> **Multi-Producer Data Parallelization**
>
> This is indeed an excellent observation, and our practical solution is straightforward. We constructed an **internal parallel server system** to solve this production-consumption imbalance. Specifically:
>
> *   We set the buffer size for each annotation server GPU to 6 clips
> *   For each training GPU, we provisioned **3–4 dedicated annotation server GPUs** to generate data
> *   This **multi-producer data parallelization** approach directly addresses the fundamental imbalance where consumption demand far exceeds production capacity
>
> By scaling the number of producers, we ensure that multiple annotation servers work in parallel to generate sufficient annotated data, preventing any bottleneck during training. This pragmatic solution proved effective throughout our 50k training steps, maintaining stable data availability without requiring complex adaptive buffer management.
>
> ---
>
> **Question 3: Cross-Modal Noise Correlation Visualization and INSS**
>
> ---
>
> Thank you for this request. The Independent Noise Sampling Strategy (INSS) is indeed a crucial technical insight, and we provide comprehensive explanations below.
>
> ### Core Problem: Spurious Structural Correlation in Sequential Noise Sampling
>
> In standard multimodal diffusion training with a single fixed random seed, noise tensors for video $\mathbf{z}_v$ and audio $\mathbf{z}_a$ are sampled sequentially from the same deterministic PRNG. Due to the Linear Congruential nature of standard PRNG algorithms, this creates a spurious structural correlation expressed as:
>
> $$
> \mathbf{z}_a \approx f(\mathbf{z}_v)
> $$
>
> During training, the model inadvertently learns this correlation as a "shortcut." Consequently, during inference, any alteration to video properties (e.g., resolution, duration) shifts the PRNG state, altering the subsequently sampled audio noise and breaking the learned correlation. This mismatch causes severe audio generation quality degradation.
>
> ### Explanation of the Simulation Logic
>
> To empirically validate the "cross-modal noise correlation" phenomenon identified in Section 3.3, we implemented a simulation that isolates the behavior of PRNGs during multi-modal sampling:
>
> 1.  **The PRNG Model (Linear Congruential Generator):**
>     We explicitly implement an LCG defined by $X_{n+1} = (aX_n + c) \mod m$. We selected this method because LCGs fundamentally expose the deterministic nature of random sampling. For visualization, we utilize low-modulus parameters ($m=2048$) to exaggerate the spectral lattice effect, making the subtle mathematical correlations visible.
>
> 2.  **Simulation of Sequential Sampling (Baseline):**
>     To mimic the standard training pipeline with a single global seed, we initialize a single generator (e.g., `seed=42`) and draw a continuous sequence. We assign Video Noise ($\epsilon_v$) and Audio Noise ($\epsilon_a$) sequentially: $\epsilon_v = r_i$ and $\epsilon_a = r_{i+1}$.
>
>     This results in a distinct **lattice structure**, visually demonstrating that $\epsilon_a$ is functionally dependent on $\epsilon_v$. The model overfits to this grid structure, causing degradation when inference conditions shift.
>
> 3.  **Simulation of Independent Noise Sampling (INSS):**
>     We instantiate two separate generators with distinct seeds (e.g., `seed_video=42`, `seed_audio=999`). Consequently, $\epsilon_v$ and $\epsilon_a$ are sampled from completely disjoint states.
>
>     The lattice structure **disappears completely**, becoming a uniform scatter cloud. This proves that $\epsilon_v$ and $\epsilon_a$ are statistically independent, ensuring the model learns semantic alignment rather than memorizing PRNG artifacts.
>
> The visualization results are provided in Fig.8 in supplementary of revision.

---

> ### Author Response · Authors · 2025-11-24
> **Response to Reviewer MU97 -- Part3**
>
> **Question 4: Audio-Video Alignment (AV-A) Metric Clarification**
>
> ---
>
> Thank you for this important clarification question. We provide precise definitions of the AV-A metric for different evaluation scenarios below.
>
> ### Core Definition: AV-A Metric
>
> The Audio-Video Alignment (AV-A) metric measures the temporal synchronization between audio and video by computing the offset distance between their Synchformer features:
>
> $$
> \text{AV-A} = \text{Synchformer}(\mathbf{v}\_{\text{gen}}, \mathbf{a}\_{\text{gen}})
> $$
>
> where $\mathbf{v}\_{\text{gen}}$ and $\mathbf{a}\_{\text{gen}}$ denote the generated video and audio modalities respectively. The metric outputs a temporal offset value (in seconds), with **lower values indicating better synchronization**.
>
> ### Critical Task-Specific Distinctions
>
> **For Joint Audio-Video Generation (UniVerse-1, SVG):**
>
> $$
> \text{AV-A}_{\text{joint}} = \text{Synchformer}(\mathbf{v}\_{gen}, \mathbf{a}\_{gen})
> $$
>
> Both modalities are generated by the model simultaneously. **No ground truth is used as reference**. This measures the model's intrinsic ability to maintain temporal coherence between two independently generated streams.
>
> **For Video-to-Audio Generation (HunyuanVideo-Foley):**
>
> $$
> \text{AV-A}_{\text{V2A}} = \text{Synchformer}(\mathbf{v}\_{gt}, \mathbf{a}\_{gen})
> $$
>
> Video is **ground truth** from the test set; only audio is generated. The model benefits from perfect visual event timing and can condition audio generation directly on known visual cues.
>
> ### Holistic Assessment: AV-A + CS
>
> The paper correctly notes that AV-A alone is insufficient for evaluating joint generation. Audio-video synchronization is only meaningful if the audio content is semantically relevant. Therefore, we report:
>
> $$
> \text{Joint Quality} = \text{Harmonic Mean}(\text{Normalized}(\text{AV-A}), \text{Normalized}(\text{CS}))
> $$
>
> where CS (CLAP score) measures semantic consistency with the text prompt. This jointly enforces both temporal alignment and content relevance:
>
> | Model | AV-A | CS | Joint Score |
> | :--- | :--- | :--- | :--- |
> | Ours | 0.23 | 0.16 | Strong balance |
> | SVG | 0.09 | 0.08 | Poor content |
>
> Our model demonstrates better overall audio-visual content consistency than SVG precisely because we optimize for both dimensions simultaneously.

---

### Meta-Review · Area_Chair_6Qbd · 2026-01-04

**Summary:**

In the initial reviews, 2 reviewers recommend acceptance and 2 recommend rejection.

MU97 is in favor of acceptance. They praise the originality of SoE and how it can be used for to reuse pretrained single-modality models. They also praise the online data annotation pipeline. They are concerned with the lack of direct comparison to Wan-1.3B. In the rebuttal, the authors point out that such a comparison is not straightforward, due to the lack of frame conditioning (indeed, only the text-to-video variant appears to be available). They also complain about the emphasis on unimodal metrics. CDjj favors acceptance, praising the writing, dataset, and code/weights that will be released. They complain about the low quality of the generated examples provided in the supplemental material and complain about the lack of intuition and novelty (in comparison to AVLink). The rebuttal outlines differences to AVLink and suggests that the shortcomings of the qualitative results are partly due to the use of the Ace-step backbone as a music generation model, which was not trained for all of the relevant audio. Gfef is in favor of rejection. They are concerned about the lack of detail about the data annotation pipeline, the lack of dataset ablations, the motivation for the independent noise sampling strategy. The rebuttal addresses the noise sampling concerns. However, the AC agrees that this issue was not clearly explained in the paper and feels that this part requires revision. The reviewer posted a comment indicating further questions about this. After extensive back-and-forth, the reviewer had remaining questions. They also raise concerns about the novelty of the AV alignment metric, which the rebuttal acknowledges and will cite. Reviewer qTAG recommended rejection. They have concerns about the description of prior work, which they feel was not credited adequately in the initial version, the limited comparison (citing the fact that they are only comparing to newly proposed Verse-Bench), the claims about being the first open source generation method, the lack of extensive evaluation of other stitching approaches, and lack of qualitative and quantitative comparisons. In the rebuttal, the authors point out that they use FD and claim that FVD is not well suited to their domain. They also point out that the cited joint generation methods do not have publicly available open source models.

The AC feels that the benefits of the approach (raised by MU97 and CDjj) are outweighted by the concerns raised by several of the reviewers, especially the issues around the qualitative results and comparisons (CDjj and qTAG) and the lack of fair comparisons to alternative joint generation and fusion methods. The AC therefore believes that the paper requires major revision before it can be accepted at ICLR.

**Reviewer Concerns:**

For MU97: the authors use reported benchmark results to illustrate the gap between the 1.3B and 14B. models. The authors are correct that a fair comparison would be challenging, though it is unclear if this (indirect) comparison would be satisfying for the reviewer. CDjj's complaints about comparisons to AVLink are addressed by the authors through a breakdown of the differences, emphasizing the joint generation capabilities.  I assume that this may partially address the complaint. For the point about qualitative results, I do not think that the author's response (which attributes the poor result sto the the lack of relevant pretraining data) would be likely to change CDjj's opinion about the quality, even if it is correct. Gfef's question about the noise sampling strategy would likely be addressed by the author response, which clarifies the issue. However, the AC agrees that the explanation is inadequate. After extensive back-and-forth, the reviewer still had questions, though these may be partially resolved by the final comment. qTAG's point about open source models and metrics was partly addressed after the clarification. However, their concerns about fusion ablations and qualitative results may not have been fully addressed by the rebuttal.

**Reviewer Scores:**

The reviewers in favor of acceptance (MU97 and CDjj) likely would have at least maintained their scores. I see no reason to believe that they would have lowered them. It is unclear how Gfef would have adjusted their score: while the rebuttal partially addresses their concerns, after an extensive back and forth, they still expressed confusion with the method. The rebuttal does not address qTAG's major concerns about the quality of the results. The AC does not see evidence that qTAG would have raised or lowered their score.

---

### Decision · Program_Chairs · 2026-01-26

Reject